# Physiological activation of Aryl hydrocarbon receptor by food-derived ligands is essential for the efficacy of anti-PD1 therapy

Alba De Juan [1,7], Alice Coillard[1,7], Adeline Cros[1], Alessandra Rigamonti[1], Lamine Alaoui[1], Julio L. Sampaio[2], Natacha Monot[1], Aurélie Balvay[3], Anne Foussier[3], Mathilde Rieux-Laucat[1], Léa Guyonnet[4], Sylvie Rabot [3], Christian Vosshenrich [5] & Elodie Segura [1,6] ✉

Cancer immuno-surveillance and response to therapy are affected by environmental factors, including nutrition. However, the direct effects of individual nutrients remain poorly understood. Here we investigate the impact of dietary ligands of Aryl hydrocarbon receptor (AhR), a transcription factor activated by tryptophan catabolites generated through food digestion and microbiota metabolism. By analyzing pre-clinical tumor models in mice fed on a diet naturally poor in AhR ligands or the same diet supplemented with Indole-3-carbinol, we show that diet-derived AhR ligands are required for the optimal efficacy of anti-PD1 therapy. Using conditional knockout mice, we evidence an essential role for AhR in CD8 T cells, but not NK cells or myeloid cells. Mechanistically, AhR promotes anti-PD1-mediated reinvigoration of progenitor exhausted CD8 T cells and licences the functional response of effector CD8 T cells. Our work allows a better understanding of the role of nutrients in anti-tumor immune responses and has implications for the rational design of dietary interventions for improving the efficacy of checkpoint blockade therapy.

Immune responses against tumors and response to checkpoint blockade therapy can be affected by multiple factors, including environment and lifestyle[1,2]. In particular, nutrition could represent an important player[3–6] but its influence on anti-tumor immune responses remains understudied.

Nutrition can modulate the immune system through metabolites, either produced by host digestion or by microbiota metabolism[6,7]. Microbiota diversity and composition has been shown to play an essential role in the efficacy of checkpoint blockade therapies[8]. Recent studies in mice models have also evidenced a role in anti-tumor immune responses specifically for dietary nutrients. Vitamin C can modulate the recruitment into tumors of T cells and their functional

properties, and high doses of Vitamin C enhance checkpoint blockade therapy[9,10]. Vitamin E was also shown to increase the efficacy of anti-PD1/PDL1 treatment by favoring the antigen presenting function of dendritic cells (DC)[11]. In addition, ketone body 3-hydroxybutyrate augments anti-tumoral immune responses by limiting the expression of inhibitory receptors and synergizes with anti-PD1 therapy[12]. These results suggest that dietary interventions could provide clinical benefits to patients being treated with checkpoint blockade therapy. However, a major hurdle is our limited knowledge of the direct effects of individual dietary nutrients on anti-tumoral immune responses.

One of the molecular links between diet and immune cells is the Aryl hydrocarbon receptor (AhR). AhR is a ligand-activated

[1]Institut Curie, PSL University, INSERM, U932, Paris, France. [2]Institut Curie, PSL University, Plateforme de Métabolomique et Lipidomique, Paris, France. [3]Université Paris-Saclay, INRAE, AgroParisTech, Micalis Institute, Jouy-en-Josas, France. [4]Institut Curie, PSL University, Plateforme de Cytométrie de Flux, Paris, France. [5]Immunology Department, Institut Pasteur, INSERM, U1223, Innate Immunity Unit, Paris, France. [6]Institut Necker Enfants Malades, INSERM, U1151, Paris, France. [7]These authors contributed equally: Alba De Juan, Alice Coillard. ✉e-mail: elodie.segura@inserm.fr

transcription factor sensing environmental signals, in particular indoles and tryptophan catabolites generated through food intake and microbiota metabolism[13]. Imbalance in gut-derived AhR ligands affects intestinal homeostasis and favors intestinal inflammatory symptoms[14–18]. However, gut-derived AhR ligands can also disseminate systemically in blood and organs[19,20], and several studies have evidenced a role for dietary AhR agonists in distant tissues. In the brain, nutritional AhR ligands limit neuro-inflammation by acting on astrocytes and microglia[21,22]. We previously showed that diet-derived AhR agonists impact steady-state monocyte differentiation in vivo in mouse skin and peritoneum[23], and control the inflammatory state of epidermal cells thereby regulating the severity of cutaneous allergic responses[24]. Dietary AhR ligands were also shown to act on lung cells, including epithelium, endothelium and immune cells[25]. These observations suggest that dietary AhR ligands may have an underappreciated role in anti-tumoral immune responses. In particular, whether nutritional AhR agonists affect immune responses upon checkpoint blockade therapy remains unknown.

In this work, we demonstrate a non-redundant role for dietary AhR agonists in promoting anti-tumoral immune responses upon anti-PD1 treatment in pre-clinical models, while imbalance in microbiota-derived ligands has no detectable impact. We find that diet-derived AhR ligands do not significantly impact the tumor-infiltrating immune cells profile at baseline, but favor reinvigoration of CD8 T cells upon anti-PD1 treatment and modulate their cross-talk with NK cells. These results shed new light on the role of dietary nutrients in anti-tumor immune responses and should have important implications for the design of dietary interventions for improving the efficacy of checkpoint blockade therapy.

## Results

### Dietary AhR ligands are essential for the efficacy of anti-PD1 immunotherapy

Analysis of mouse serum showed that standard chow diet naturally contains the AhR pro-ligand indole-3-carbinol (I3C) (Supplementary Fig. 1a), which can be transformed by the gastric acidic milieu into potent AhR agonists[26,27]. To study the impact of dietary AhR agonists on anti-tumoral immune responses, we therefore used a purified diet naturally poor in phytochemicals (hereafter termed 'indole-poor diet') and compared with the same diet complemented with I3C (hereafter termed 'I3C diet'). In this setting, mice fed on I3C diet display physiological levels of I3C in the serum, comparable to mice fed on a standard chow diet, and circulating levels of I3C (but not other indole derivatives) are significantly decreased in the indole-poor diet group as previously reported[24] (Supplementary Fig. 1a). Mice fed on these two diets have comparable weight after 3 weeks of adaptation (Supplementary Fig. 1b). To rule out a possible effect of the indole-poor diet on intestinal barrier integrity[28], which could impact the efficacy of anti-PD1 therapy via microbiota translocation[29], we assessed intestinal permeability by measuring diffusion of fluorescent dextran to the blood. There was no difference between mice fed on I3C or indole-poor diet (Supplementary Fig. 1c), indicating that the intestinal barrier was not compromised by the lack of dietary AhR ligands. To confirm that indole-poor and I3C diets induced different levels of AhR activation in vivo, we analyzed the expression in the small intestine of the canonical AhR target gene *Cyp1a1* (Supplementary Fig. 1d). Mice fed on the indole-poor diet had undetectable expression of *Cyp1a1* in the duodenum, where AhR ligands mainly come from the stomach, while *Cyp1a1* expression was similar between diet groups in the colon, where AhR ligands mostly come from colon microbiota. These observations confirm that the indole-poor diet contains negligeable amounts of natural AhR ligands.

To address whether dietary AhR agonists modulate the efficacy of checkpoint blockade therapy, we used fibrosarcoma tumor cells expressing the model antigen Ovalbumin (MCA101-OVA), and assessed tumor growth in mice treated or not with anti-PD1. Treatment consisted of 3 injections, starting when the tumor volume reached 80–100 mm³. These conditions induced heterogenous responses: no observable effect ('tumor progression'), tumor regression or stabilization during a minimum of 10 days before progressing again (classified as 'partial response') or tumor rejection, mirroring the situation observed in patients treated with checkpoint blockade therapy[30]. Anti-PD1 treatment limited tumor progression in a majority of mice fed on I3C diet (Fig. 1a, b). However, there was limited efficacy of anti-PD1 treatment in mice fed on indole-poor diet (Fig. 1a, b), displaying at day 10 of therapy overall larger tumors (Fig. 1c) and smaller change in tumor volume compared to baseline (Fig. 1d). By contrast, there was no impact of diet in the absence of anti-PD1 treatment on tumor progression (Fig. 1a, b) or growth kinetics (Fig. 1c, d), suggesting that the diet had an impact on the immune system rather than directly on tumor cells. Moreover, we found no evidence for AhR activation in MCA101-OVA cells when exposed in vitro to AhR agonists (Supplementary Fig. 1e), indicating that this tumor model does not respond to AhR ligands directly. To confirm the involvement of immune cells, we rechallenged with MCA101-OVA cells the mice fed on I3C diet which had rejected the tumor, using naive mice fed on I3C diet as control for tumor growth. Tumors were quickly rejected in all rechallenged mice (Fig. 1e), indicating that immune memory had been developed against tumor cells. Finally, to confirm that these observations were not restricted to the pre-clinical fibrosarcoma model, we performed similar experiments using the E0771 breast tumor (Supplementary Fig. 1f, g) and B16F10 melanoma (Supplementary Fig. 1h, i) models. Mice fed on indole-poor diet were less responsive to treatment also in these models. These results indicate that lack of dietary AhR ligands is detrimental for the optimal efficacy of anti-PD1 treatment.

### Decreased production of microbiota-derived AhR ligands has no detectable impact on the efficacy of anti-PD1 immunotherapy

AhR ligands produced from microbiota or endogenous metabolism have been reported to influence anti-tumor immune responses[31–35]. To determine whether lack of dietary AhR ligands could modify in the tumor the abundance of AhR ligands from other sources, we measured several metabolites in tumor lysates using HPLC-HRMS. I3C relative abundance was severely reduced in tumors of mice fed on indole-poor diet (Fig. 2a), indicating that food-derived ligands can reach the tumor milieu. By contrast, the relative abundance of L-kynurenine (host-derived[36] or microbiota-derived[37,38]), Indole-3-acetic acid and Indole-3-lactic acid (host-derived[33] or microbiota-derived[39,40]), and Indole-3-propionic acid, indole-3-carboxyaldehyde and Indoxyl sulfate (microbiota-derived[19]) was not impacted by the diet (Fig. 2a). These results suggest a specific role for food-derived AhR ligands in anti-PD1 therapy, that cannot be compensated for by ligands from other sources.

To directly address the impact of microbiota-derived AhR agonists in our model, we generated mice in which the microbiota is selectively impaired for the production of indole-derived AhR ligands through tryptophan catabolism. We used germ-free mice, fed on standard chow diet, reconstituted with tryptophanase-sufficient (wild-type, WT) or -deficient (KO) *E. coli*. Bacteria implantation was similar in both groups (Supplementary Fig. 2a). Only mice reconstituted with WT, but not with KO, *E. Coli* displayed indole in the feces (Supplementary Fig. 2b), indicating efficient production of indoles in mice reconstituted with WT *E. Coli*. We then assessed the relative abundance of other AhR ligands in the serum. Mice reconstituted with KO *E. Coli* had significantly lower levels of circulating indoxyl sulfate and indole-3-propionic acid (Supplementary Fig. 2c), which are both indole derivatives[19], compared to mice reconstituted with WT *E. Coli*. By contrast, other ligands including I3C showed similar abundance between groups. Of note, anti-PD1 treatment in these mice was not as efficient as in conventional mice (Fig. 2b–d), which is likely due to the altered development and homeostasis of the immune system in the

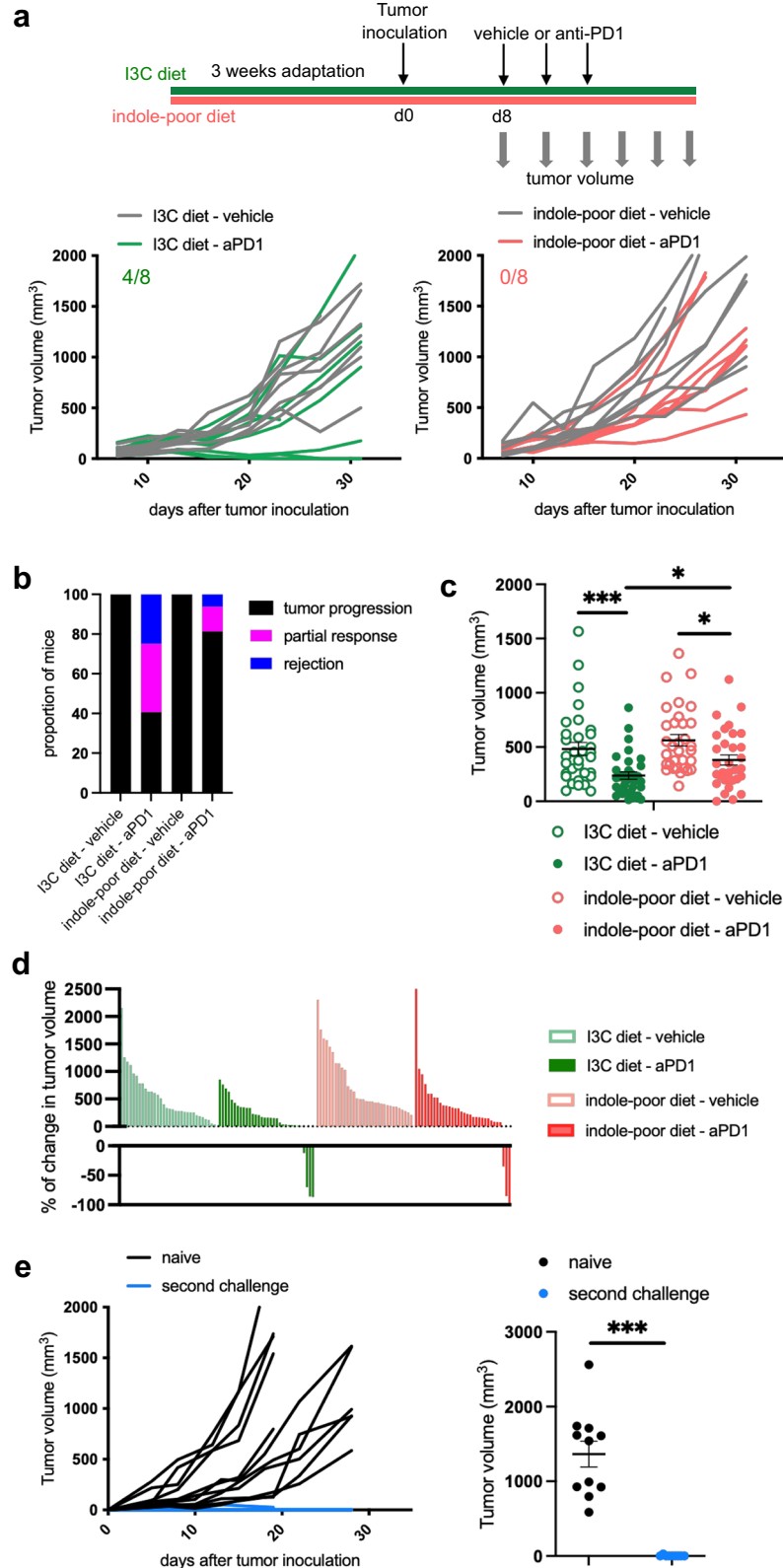

**Fig. 1 | Dietary AhR ligands are essential for the efficacy of anti-PD1 immunotherapy in a fibrosarcoma preclinical model.** Mice were inoculated with MCA101-OVA tumor cells. When tumors reached 80–100 mm³, mice were treated intra-peritoneally with 3 doses of anti-PD1 or vehicle. **a–d** Mice were placed on Indole-poor diet or I3C diet for 3 weeks of adaptation prior to the start of experiments. **a** Representative tumor growth kinetics (1 cohort out of 4). Each line represents an individual mouse (*n* = 8). **b** Proportion of mice showing tumor rejection, partial response, or tumor progression (*n* = 32 in 4 cohorts). **c** Tumor volume at day 10 after the first treatment dose (*n* = 32 in 4 cohorts). One-way

ANOVA. **d** Percentage of change in volume for each tumor between day 0 and day 10 of treatment (*n* = 32 in 4 cohorts). **e** Mice (fed on I3C diet) which had rejected the tumor were rechallenged with MCA101-OVA cells. Tumor growth was compared to naive mice injected with the tumor at the same time. Tumor growth kinetics. Tumor volume at endpoint. Mean +/− sem are shown (*n* = 9–11 in 3 cohorts). Two-tailed Two-tailed Mann–Whitney test. For all panels, *$p < 0.05$, **$p < 0.01$, ***$p < 0.001$. Absence of star indicates 'not significant'. Source data are provided as a Source Data file.

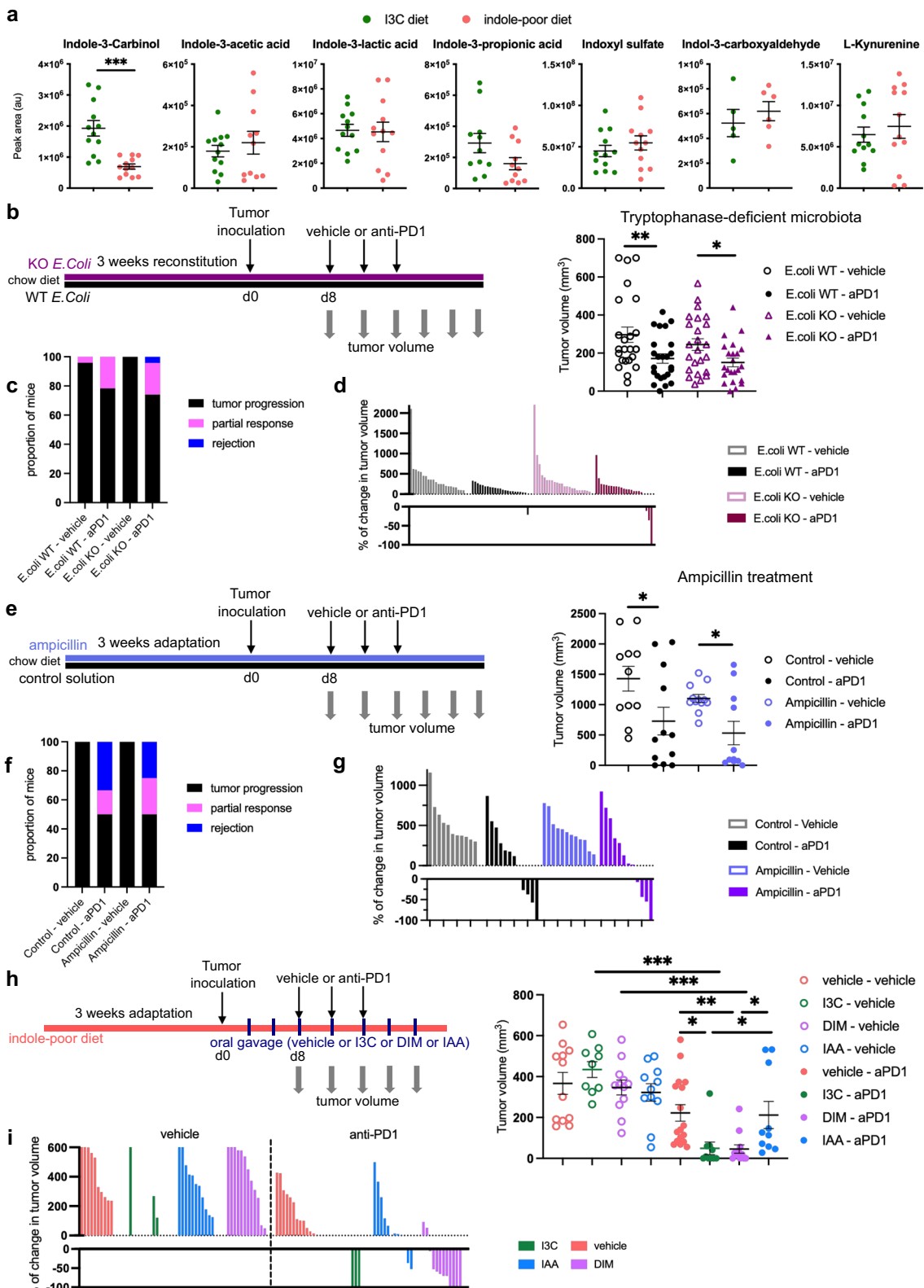

absence of microbiota[41]. Nevertheless, there was no significant difference in tumor growth or response to treatment between mice harboring WT or KO *E. coli* (Fig. 2b–d), indicating that indoxyl sulfate and indole-3-propionic acid are dispensable for the efficacy of anti-PD1 therapy. To test the role of other microbiota-derived AhR ligands, we depleted conventional mice (specific pathogen-free, fed on standard chow diet) of ampicillin-sensitive species, among which *Lactobacilli*

are known to produce AhR ligands indole-3-lactic acid, indole-3-carboxyaldehyde and indole-3-propionic acid[19,39]. Consistent with this, the relative abundance of these metabolites was lower in ampicillin-treated mice compared to control group (Supplementary Fig. 2d). However, ampicillin treatment had no impact on anti-PD1 efficacy (Fig. 2e–g), suggesting that these microbiota-derived ligands were dispensable. Finally, in mice fed on indole-poor diet, we directly

**Fig. 2 | Imbalance in microbiota-derived AhR ligands has no detectable impact on the efficacy of anti-PD1 therapy. a** Mice were placed on Indole-poor diet or I3C diet for 3 weeks of adaptation. Mice were inoculated with MCA101-OVA tumor cells. Tumors were analyzed when they reached 80–100 mm$^3$. Relative amount of metabolites was measured in tumor lysates at day 8 after tumor inoculation. Mean +/− sem are shown (*n* = 5–12 biological replicates in 2–3 independent experiments). Two-tailed Mann–Whitney test. **b–g** Mice were fed on standard chow diet. **b–d** Germ-free mice were reconstituted with Tryptophanase-competent (WT) or -deficient (KO) *E. coli*. **b** Tumor volume at day 10 after the first treatment dose (*n* = 23 in 3 cohorts). One-way ANOVA. **c** Proportion of mice showing tumor rejection, partial response, or tumor progression (*n* = 23 in 3 cohorts). **d** Percentage of change in volume for each tumor between day 0 and day 10 of treatment (*n* = 23 in 3 cohorts). **e–g** Mice were treated orally with Ampicillin or vehicle control.

**e** Tumor volume at day 10 after the first treatment dose (*n* = 11–12 in 2 cohorts). One-way ANOVA. **f** Proportion of mice showing tumor rejection, partial response, or tumor progression (*n* = 11–12 in 2 cohorts). **g** Percentage of change in volume for each tumor between day 0 and day 10 of treatment (*n* = 11–12 in 2 cohorts). **h, i** Mice were placed on Indole-poor diet for 3 weeks of adaptation. Mice were inoculated with MCA101-OVA tumor cells and supplemented 3 times a week by oral gavage with I3C, 3,3′-Diindolylmethane (DIM) or Indole-3-acetic acid (IAA). When tumors reached 80–100 mm$^3$, mice were treated intra-peritoneally with 3 doses of anti-PD1 or vehicle. **h** Tumor volume at day 10 after the first treatment dose (*n* = 9–17 in 2–4 cohorts). One-way ANOVA. **i** Percentage of change in volume for each tumor between day 0 and day 10 of treatment (*n* = 9–17 in 2–4 cohorts). For all panels, *$p < 0.05$, **$p < 0.01$, ***$p < 0.001$. Absence of star indicates 'not significant'. Source data are provided as a Source Data file.

compared oral supplementation with I3C, I3C-derived agonist DIM or Indole-3-acetic acid (IAA) which is mainly produced from microbiota metabolism. Response to anti-PD1 treatment was significantly more efficient in mice fed on indole-poor diet and supplemented with I3C or DIM, but not in mice fed on indole-poor diet and supplemented with IAA (Fig. 2h, i). Collectively, these results show that decreased production of AhR agonists derived from microbiota metabolism has no major impact on the efficacy of anti-PD1 therapy.

## Lack of dietary AhR ligands does not significantly affect the baseline tumor myeloid profile

To decipher the mechanisms involved, we first analyzed the tumor micro-environment prior to therapy. We profiled by single-cell RNA sequencing (scRNA-seq) the tumor-infiltrating mononuclear phagocytes (CD11b$^{high}$ cells) in tumors of 80–100 mm$^3$ volume. Using unsupervised clustering, we identified populations of monocytes (expressing *Plac8, Ly6c2*), 3 clusters of monocyte-derived macrophages (mo-Mac) (*Plac8, Ly6c2, Lyz2*), several macrophage populations (*C1qc, C1qa, Apoe*) that can be classified into macrophages with high expression of MHC II genes, *Arg1*-high macrophages, 4 clusters of tissue macrophages (*Mrc1, Stmn1*) including a cluster with a cycling signature (*Mki67, Tubb5, Top2a*), and finally cDC2 (*Flt3, Ccr7*, MHC II genes) (Fig. 3a, b and Supplementary Fig. 3a). Comparing the proportion of these clusters between diet groups showed a tendency for increased tissue macrophages in the I3C diet group (Fig. 3c).

To validate these observations, we analyzed tumor-infiltrating myeloid cells by flow cytometry using CD206 (encoded by *Mrc1*) and Ly6C to identify tissue macrophages and mo-Mac, respectively (Supplementary Fig. 3b). CD206$^+$ tissue macrophages were more abundant in tumors of mice fed on the I3C diet (Fig. 3d). Proportions of mo-Mac (Fig. 3d), monocytes, CD206$^-$Ly6C$^-$ macrophages, cDC2, cDC1, monocyte-derived DC and neutrophils were similar between diet groups (Supplementary Fig. 3c). Collectively, these results indicate that lack of dietary AhR ligands does not modify significantly the baseline myeloid cell composition except for decreased proportions of CD206$^+$ tissue macrophages.

## AhR activation in macrophages does not affect responses to anti-PD1 therapy

We then examined the hypothesis that the absence of dietary AhR ligands could impact the transcriptional program of macrophages. In particular, mo-Mac clusters displayed high expression of *Cd244*, an inhibitory receptor proposed to suppress immune responses in tumor-associated myeloid cells[42,43]. Flow cytometry analysis confirmed that CD244 expression was higher in tumor-infiltrating mo-Mac from mice fed on indole-poor diet (Supplementary Fig. 3e), suggesting that lack of dietary AhR ligands could favor a pro-tumorigenic phenotype in macrophages. To directly test whether AhR in macrophages was involved in response to anti-PD1 therapy, we inoculated MCA101-OVA tumors to mice deficient for AhR in myeloid cells (LysM*AhR$^\Delta$ mice)

and wild-type (WT) littermates, fed on indole-poor or I3C diet. All mice were treated with anti-PD1. WT and LysM*AhR$^\Delta$ mice fed on I3C diet both responded to anti-PD1 treatment (Supplementary Fig. 3f–h), while mice fed on indole-poor diet showed limited efficacy of anti-PD1 therapy in both WT and LysM*AhR$^\Delta$ groups. We conclude that lack of AhR activation by dietary ligands specifically in macrophages does not affect the efficacy of anti-PD1 therapy in this tumor model.

## Lack of dietary AhR ligands does not significantly modify the baseline tumor T cells composition

CD8 T cells were shown to be essential for the eradication of MCA101-OVA tumors[44], therefore we next examined the T cells profile before therapy. We analyzed by scRNA-seq the tumor-infiltrating T lymphocytes (TCRβ$^+$ cells) in tumors of 80–100 mm$^3$ volume. Using unsupervised clustering, we identified populations of NK T cells (*Fce1g, Xcl1, Klra7*), effector CD8 T cells (*Cd8a, Ccl5, Gzmk*), exhausted CD8 T cells (*Cd8a, Pdcd1, Lag3*), stem-like CD8 T cells (*Cd8a, Tcf7, Lef1*), progenitor exhausted T cells (*Tcf7, Slamf6*), IFN-responsive cells (*Ifit3, Ifit1, Isg15*), cycling T cells (*Stmn1, Top2a, Mki67*) and 2 clusters of T regulatory cells (Treg) (*Foxp3, Il2ra, Itgae, Ctla4*) including one expressing higher levels of *Lag3, Gzmb* and *Tnfrsf18* (encoding GITR) (Fig. 3e, f and Supplementary Fig. 4a). Proportions of each cluster between diet groups were overall similar (Fig. 3g), with a tendency for higher Treg proportions in tumors from the I3C diet group. To confirm these results, we analyzed tumor-infiltrating lymphoid cells using flow cytometry (Supplementary Fig. 4b). There was no significant difference between diet groups in the proportions of CD4 T cells and Treg, and of CD8 T cells, including exhausted cells (Supplementary Fig. 4c and Fig. 3h). The proportions of NK cells and NKT cells were also similar (Supplementary Fig. 4c). These observations indicate that lack of dietary AhR ligands does not modify the baseline T cells profile in tumors.

Finally, we assessed the expression of PD1 on CD8 T cells and NK cells. The proportion of PD1$^+$ CD8 T cells was higher in tumors of mice fed on the I3C diet (Supplementary Fig. 4d), as well as the expression level in PD1-positive cells (Fig. 3i). This difference may be relevant for the efficacy of therapy, as reactivity to anti-PD1 treatment correlates with PD1 expression levels on CD8 T cells in some cancers[45,46]. NK cells did not express PD1 in this tumor model, irrespective of the diet (Supplementary Fig. 4e).

Collectively, these results show that lack of dietary AhR ligands does not affect the proportion of tumor-infiltrating total or exhausted CD8 T cells, but potentially impacts their reactivity to treatment by down-modulating PD1 surface expression.

## Dietary AhR ligands do not impact the induction of antigen-specific immune responses

Because the pre-existence of tumor-infiltrating CD8 T cells favors responses to anti-PD1 treatment[46], we then examined the effect of dietary AhR ligands on the induction of antigen-specific CD8 T cell responses at baseline. Using flow cytometry, we first assessed DC

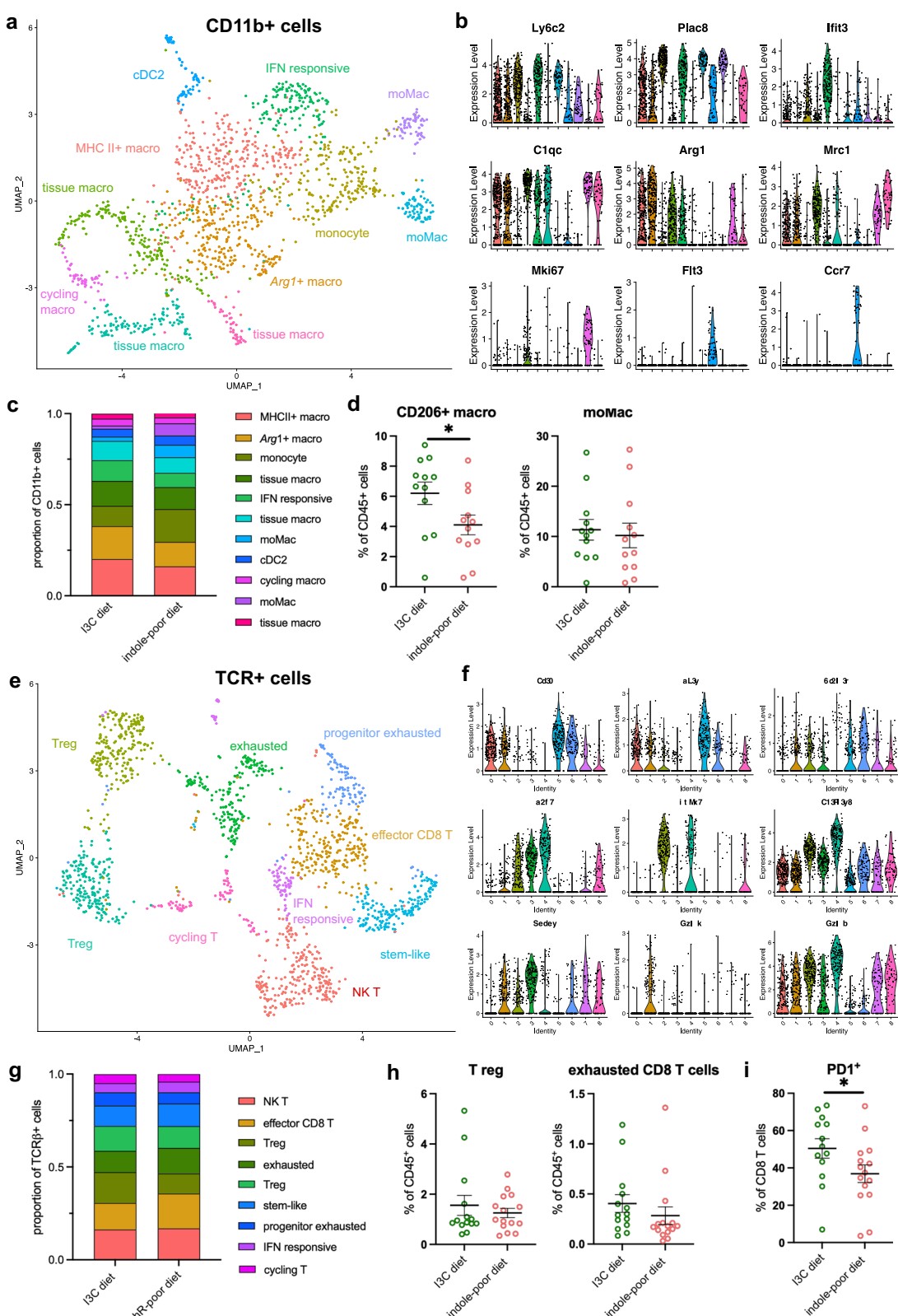

migration to tumor-draining lymph nodes, a pre-requisite for the induction of antigen-specific responses. There was no impact of the diet on resident or migratory DC proportions (Supplementary Fig. 5a), suggesting that dietary AhR ligands do not influence DC migration to lymph nodes. We then assessed the presence in tumor-draining lymph nodes (Supplementary Fig. 5b, c) and in tumors (Supplementary Fig. 5d) of OVA-specific CD8 T cells using tetramer staining. There was

no difference between diet groups in the proportion of CD8 T cells expressing the OVA-specific T cell receptor prior to therapy. Finally, we addressed whether lack of dietary AhR ligands impairs antigen-specific responses during anti-PD1 therapy in tumor-draining lymph nodes. We measured OVA-specific CD8 T cells 48 h after anti-PD1 injection, using tetramer staining (Supplementary Fig. 5e). We found no impact of the diet on antigen-specific CD8 T cells. Taken together, these

**Fig. 3 | Lack of dietary AhR ligands does not affect the baseline tumor-infiltrating immune profile.** Mice were placed on Indole-poor diet or I3C dietfor 3 weeks of adaptation prior to the start of experiments. Mice were inoculated with MCA101-OVA tumor cells. Tumors were analyzed when they reached 80–100 mm³. **a–c** Tumor-infiltrating CD11b+ cells were analyzed by scRNA-seq (n = 2 biological replicates per group). **a** UMAP representation of clusters identified. **b** Expression of marker genes across clusters. **c** Proportion of cells belonging to each cluster across diet groups. **d** Cells were analyzed by flow cytometry. Proportion of indicated populations among live CD45+ cells. Mean +/− sem are shown (n = 12 biological replicates in 2 independent experiments). Two-tailed Mann–Whitney test.

**e–g** Tumor-infiltrating TCRb+ cells were analyzed by scRNA-seq (n = 2 biological replicates per group). **e** UMAP representation of clusters identified. **f** Expression of marker genes across clusters. **g** Proportion of cells belonging to each cluster across diet groups. **h, i** Cells were analyzed by flow cytometry. **h** Proportion of indicated populations among live CD45+ cells. Mean +/− sem are shown (n = 14–15 biological replicates in 3 independent experiments). **i** Proportion of CD8 T cells expressing PD1. Mean +/− sem are shown (n = 14–15 biological replicates in 3 independent experiments). Two-tailed Mann–Whitney test. For all panels, *p < 0.05. Absence of star indicates 'not significant'. Source data are provided as a Source Data file.

observations suggest that the induction and infiltration of tumor antigen-specific CD8 T cells is not affected by the lack of dietary AhR ligands.

## Lack of dietary AhR ligands impairs CD8 T cells reinvigoration and NK cells infiltration following anti-PD1 treatment

Based on the above results, we hypothesized that dietary AhR ligands modulate the efficacy of anti-PD1 treatment by acting on the response to anti-PD1 rather than on the baseline immune cells profile. To decipher the cellular mechanisms involved, we screened the expression of multiple chemokines, effector and suppressive molecules in the tumors of mice fed on I3C or indole-poor diet, before and 24 h after anti-PD1 injection. Expression of *Ccl5* and *Cxcl9*, two chemokines playing an essential role in CD8 T cells trafficking to tumors upon anti-PD1 treatment[47], was significantly increased in the tumors of treated mice, independently of the diet (Fig. 4a and Supplementary Fig. 6a). By contrast, the expression of *Gzmb* was upregulated upon treatment only in tumors of mice fed on I3C diet (Fig. 4a and Supplementary Fig. 6a), suggesting that dietary AhR ligands could modulate cytotoxic responses.

Anti-PD1 treatment induces the reinvigoration of CD8 T cells, including their rapid expansion[47]. To directly assess cell proliferation, we stained tumor-infiltrating cells for Ki67 (Supplementary Fig. 6b), a molecule that accumulates upon entry into cell cycle[48]. Anti-PD1 treatment significantly increased the proportion of Ki67+ CD8 T cells, but only in tumors of mice fed on the I3C diet (Fig. 4b). We next quantified CD8 T cells in tumors 48 h after treatment, but there was no significant difference in CD8 T cell numbers at this stage (Fig. 4c), suggesting that CD8 T cell expansion in the tumor requires longer time. We then analyzed by flow cytometry their production of Granzyme B and IFNγ (Fig. 4c and Supplementary Fig. 6c, d). There was no difference in the production of these effector molecules, either between vehicle and anti-PD1 or between diet groups.

We then examined natural killer (NK) cells. Anti-PD1 treatment increased the numbers of NK cells in tumors only in mice fed on I3C diet (Fig. 4d). The number, but not proportion, of NK cells producing Granzyme B and IFNγ was also increased (Fig. 4d and Supplementary Fig. 6e). However, the expression of Ki67 was not modified after anti-PD1 treatment (Supplementary Fig. 6f), indicating that recruitment of NK cells was affected rather than in situ proliferation. To examine the importance of NK cells' immune surveillance in this tumor model, we assessed growth of MCA101-OVA tumors in mice specifically lacking NK cells. To this end, we used mice deficient for the common γ chain in NK cells (Ncr1*Il2rg^Δ mice)[49] and WT littermates, fed on standard chow diet. NK cell-deficient mice displayed accelerated tumor growth with larger tumors from an early stage (Supplementary Fig. 6g). In addition, we analyzed responses to anti-PD1 therapy in mice depleted for NK cells at the same time as treatment. There was a small effect on efficacy of therapy, as shown by a trend for bigger tumors at day 10 of treatment in mice depleted for NK cells, but not statistically significant (Supplementary Fig. 6h). These observations indicate that in this model NK cells participate to

anti-tumor responses, but their absence during anti-PD1 treatment can be compensated for by other immune cells.

Collectively, these results suggest that lack of dietary AhR ligands compromises early responses to anti-PD1 treatment by impairing CD8 T cells reinvigoration and NK cells infiltration in the tumor.

## AhR is required in T cells, but dispensable in NK cells, for optimal efficacy of anti-PD1 therapy

We then sought to identify the cellular target of dietary AhR ligands among tumor-infiltrating immune cells. To determine which populations expressed AhR, we used reporter mice with TdTomato fluorescence in AhR-expressing cells[50]. Neutrophils showed no TdTomato fluorescence whether in bone marrow or tumor, while monocytes had the highest expression (Supplementary Fig. 7a and Fig. 5a). Macrophages also showed high AhR expression in tumors, as expected[31]. CD8 T cells showed TdTomato expression only in tumor (Supplementary Fig. 7a and Fig. 5b), while NK cells and CD4 T cells displayed intermediate levels of fluorescence, in both bone marrow and tumor (Supplementary Fig. 7a and Fig. 5c). In addition, CD4 T cells but not CD8 T cells expressed TdTomato in tumor-draining lymph nodes (Supplementary Fig. 7a). These observations indicate that, in addition to monocytes, macrophages, NK cells and CD4 T cells, AhR is expressed by CD8 T cells specifically in tumors. To address whether AhR was required in NK cells or T cells in a cell-intrinsic fashion, we used mice deficient for AhR in NK cells (Nrc1*AhR^Δ mice) or in T cells (Lck*AhR^Δ mice). AhR has been proposed to control NK cell functions and migration[51,52]. To assess whether AhR deficiency impacts NK cells homeostasis, we first analyzed their phenotype in the spleen. NK cells phenotype and subpopulations were similar between WT and deficient mice (Supplementary Fig. 7b). We then examined their functional properties in an ex vivo assay. AhR-deficient NK cells showed no defect in producing IFNγ or in degranulation as assessed by CD107a surface expression[53] (Supplementary Fig. 7c). To assess tumor infiltration by AhR-deficient NK cells, we inoculated MCA101-OVA tumors to Nrc1*AhR^Δ mice and WT littermates, fed on I3C diet. We quantified immune cells in tumors 48 h after treatment with vehicle or anti-PD1. Numbers of NK cells and CD8 T cells were similar between WT and deficient mice (Fig. 5d). In addition, the expression of Granzyme B and IFNγ by NK cells was not affected by AhR deficiency (Supplementary Fig. 7d), suggesting that AhR is dispensable for NK cell cytotoxic activity in vivo. These observations indicate that AhR in NK cells is dispensable for cellular responses to anti-PD1 treatment.

We then inoculated MCA101-OVA tumors to Lck*AhR^Δ mice and WT littermates, fed on I3C diet, and analyzed tumors 48 h after treatment with vehicle or anti-PD1. NK cells numbers were increased in tumors upon anti-PD1 treatment, only in WT mice (Fig. 5e). By contrast, CD8 T cell numbers were not different between WT and deficient mice (Fig. 5e). These results suggest that AhR in T cells regulates NK cells infiltration into tumors. To directly address whether AhR was required in T cells for response to anti-PD1 therapy, we assessed tumor growth in WT and Lck*AhR^Δ mice, fed on I3C diet. To exclude an indirect effect of AhR deficiency in T cells on the gut microbiota[54], WT and

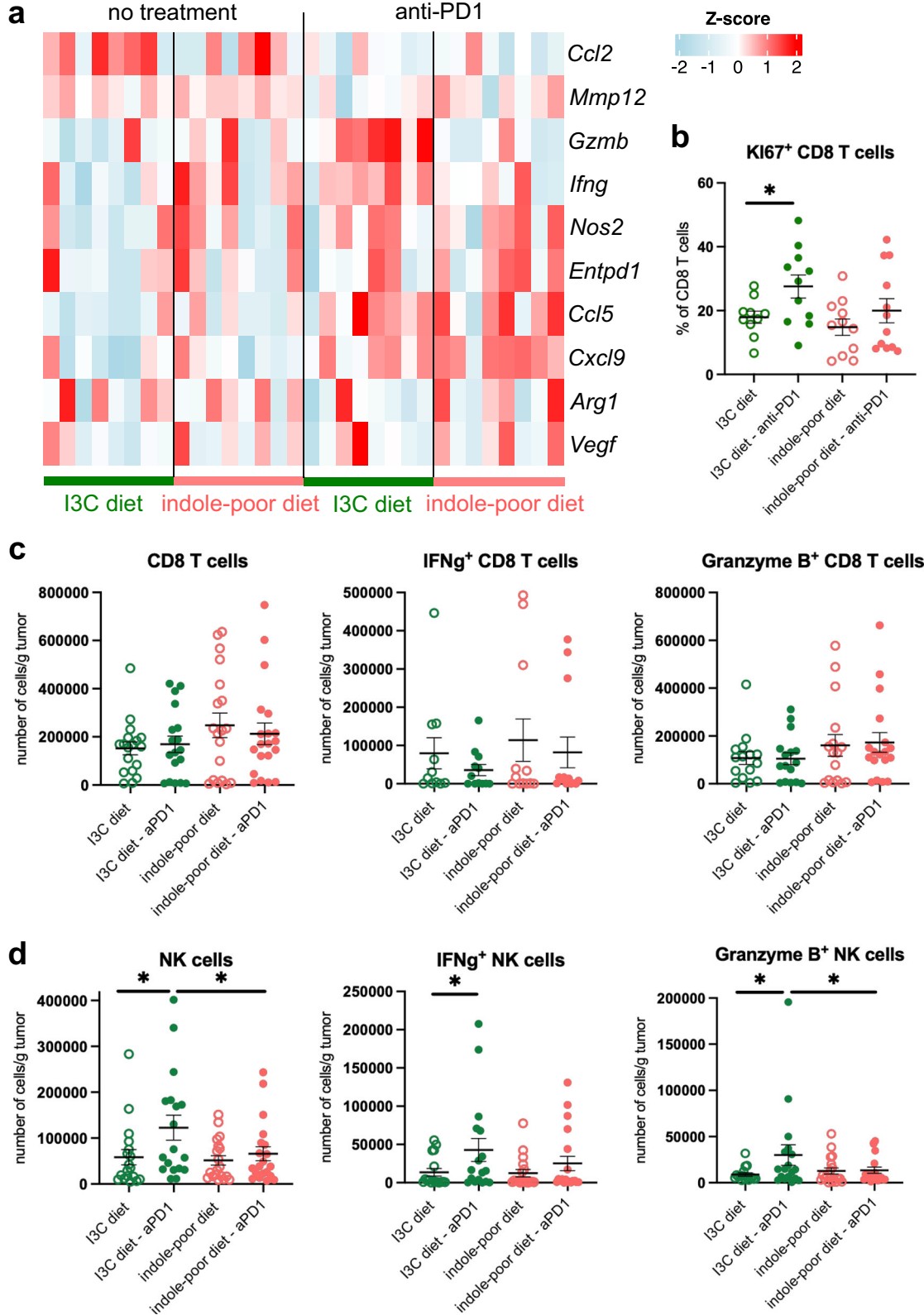

**Fig. 4 | Lack of dietary AhR ligands impairs CD8 T cells reinvigoration and NK cells infiltration following anti-PD1 treatment.** Mice were placed on Indole-poor diet or I3C dietfor 3 weeks of adaptation prior to the start of experiments. Mice were inoculated with MCA101-OVA tumor cells. Mice were treated when tumors reached 80–100 mm³. **a, b** Tumors were analyzed either on the day of injection (no treatment) or 24 h after anti-PD1 injection. **a** Tumor-infiltrating immune cells were analyzed by RT-qPCR. Scaled expression of indicated genes (*n* = 8 biological replicates). **b** Tumor-infiltrating immune cells were analyzed by flow cytometry.

Proportion of Ki67⁺ CD8 T cells. Mean +/− sem are shown (*n* = 11–12 biological replicates in 2 independent experiments). **c, d** Tumor-infiltrating immune cells were analyzed by intracellular flow cytometry, 48 h after vehicle or anti-PD1 injection. Number of indicated cell populations per gram of tumor. Mean +/− sem are shown (*n* = 18–20 in 6 cohorts). T cells **c** were analyzed after 3 h of ex vivo stimulation. NK cells (**d**) were analyzed directly after isolation. One-way ANOVA. For all panels, *$p < 0.05$. Absence of star indicates 'not significant'. Source data are provided as a Source Data file.

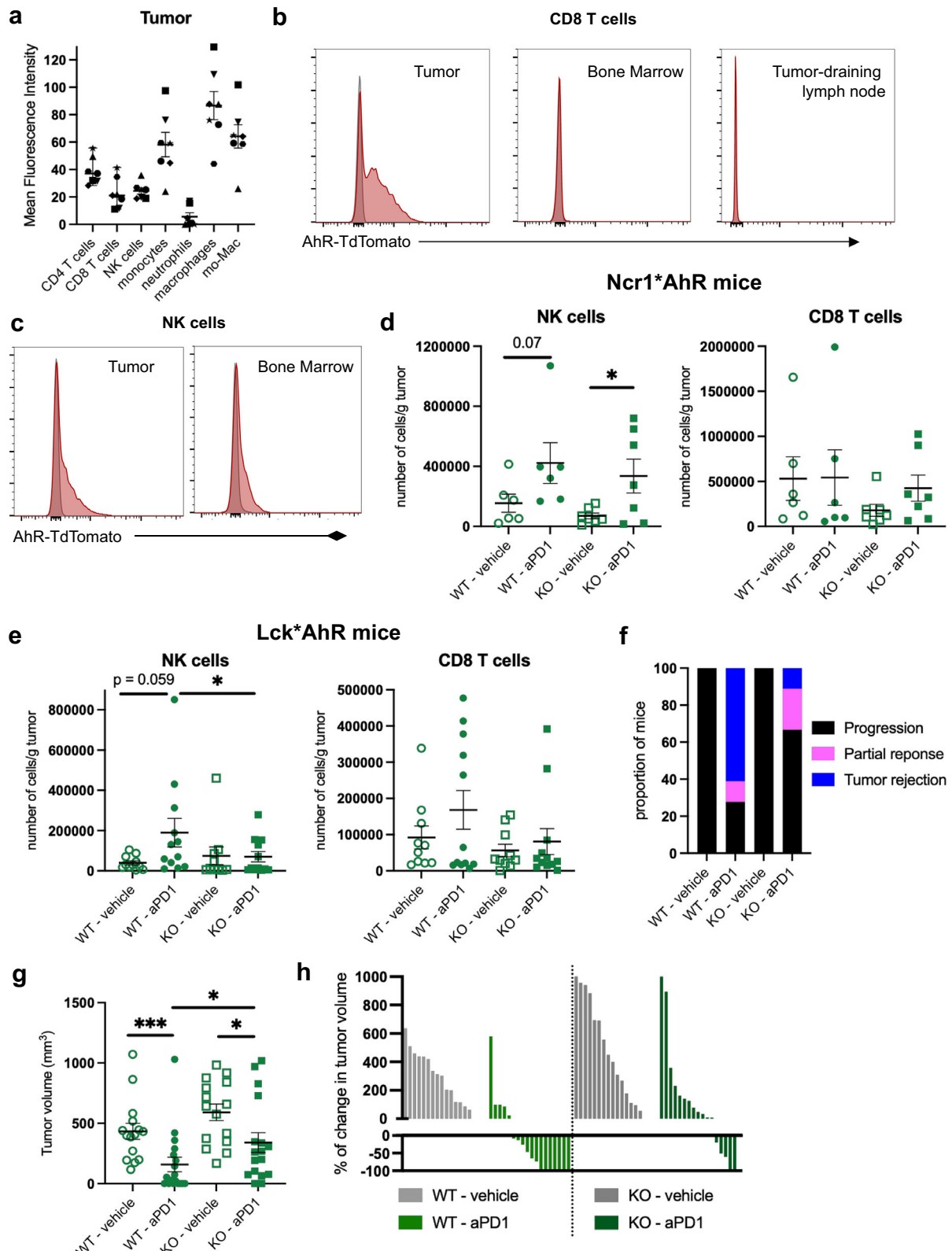

Lck*AhR^Δ mice were co-housed in the same cages in this series of

experiments. Anti-PD1 treatment was less efficient in deficient
mice (Fig. 5f–h), showing at day 10 of therapy larger tumors
(Fig. 5g) and smaller change in tumor volume compared to
baseline (Fig. 5h). We conclude that the impact of dietary AhR
ligands on the efficacy of anti-PD1 treatment primarily lies on AhR
activation in T cells.

## AhR promotes the re-invigoration of 'progenitor exhausted' CD8 T cells

We then sought to decipher what gene programs are controlled by
AhR during reinvigoration of tumor-infiltrating T cells mediated by
anti-PD1. 'Progenitor exhausted' (also referred to as 'stem-like') PD1+
Slamf6+ CD8 T cells are considered the main responder population in
anti-PD1 blockade therapy, in both mouse models and cancer

**Fig. 5 | AhR is required in T cells, but dispensable in NK cells, for optimal efficacy of anti-PD1 therapy.** Mice were inoculated with MCA101-OVA tumor cells. Mice were placed on I3C diet for 3 weeks of adaptation prior to the start of experiments. **a**–**c** AhR-TdTomato reporter mice were used, and C57Bl/6 mice for background control. Mice were analyzed when tumors reached 80–100 mm³. **a** Mean Fluorescence Intensity of TdTomato in tumor-infiltrating cell populations (n = 7 biological replicates in 2 cohorts). Each symbol represents one individual mouse. **b-c** Representative flow cytometry plots for TdTomato in CD8 T cells (**b**) or NK cells (**c**) in indicated organs. Gray histograms represent background fluorescence. **d, e** Mice were treated when tumors reached 80–100 mm³ with one dose of vehicle or anti-PD1. Tumor-infiltrating immune cells were analyzed by flow cytometry 48 h after treatment. Number of indicated cell populations per gram of tumor. **d** Ncr1*AhR^Δ mice (KO) and WT littermates were used. Mean +/− sem are shown (n = 6–7 biological replicates in 2 cohorts). Kruskall–Wallis test. **e** Lck*AhR^Δ mice (KO) and WT littermates were used. Mean +/- sem are shown (n = 10–12 biological replicates in 3 cohorts). One-way ANOVA. **f**–**h** Mice were treated when tumors reached 80–100 mm³ with 3 doses of anti-PD1 or vehicle. Lck*AhR^Δ mice (KO) and WT littermates were used. **f** Proportion of mice showing tumor rejection, partial response, or tumor progression (n = 15–18 biological replicates in 3 cohorts). **g** Tumor volume at day 10 after the first treatment dose. Mean +/- sem are shown (n = 15–18 biological replicates in 3 cohorts). Kruskall–Wallis test. **h** Percentage of change in volume for each tumor between day 0 and day 10 of treatment (n = 15–18 biological replicates in 3 cohorts). For all panels, *p < 0.05, **p < 0.01, ***p < 0.001. Absence of star indicates 'not significant'. Source data are provided as a Source Data file.

patients[45,55–58]. Therefore we performed RNA-seq analysis of tumor-infiltrating AhR-sufficient or AhR-deficient 'progenitor exhausted' CD8 T cells, 24 h after anti-PD1 treatment. We used Lck*AhR^Δ mice and WT littermates, fed on I3C diet. Differentially expressed genes were mainly enriched in WT 'progenitor exhausted' T cells (236 genes) as compared to AhR-deficient cells (10 genes) (Supplementary Data 1 and Supplementary Data 2). We then performed pathway enrichment analysis. Genes enriched in WT 'progenitor exhausted' T cells were related to mitochondrial respiration processes, with multiple instances of 'electron transport' and 'cellular respiration' pathways (Fig. 6a), in particular 'oxidative phosphorylation' (Fig. 6a, b). Other enriched pathways included cell proliferation-related pathways ('DNA repair', 'G2-M checkpoint') (Fig. 6b). Altered metabolism has been reported in dysfunctional exhausted T cells[59–62], in particular impaired mitochondrial oxidative phosphorylation[63]. Multiple genes from this pathway were more expressed in WT 'progenitor exhausted' T cells following anti-PD1 treatment (Fig. 6c), suggesting that AhR was required for optimal metabolic rewiring. Other genes potentially involved in T cell reinvigoration were upregulated in WT cells, including immune mediators and genes related to cell cycle and proliferation (Fig. 6d). Of note, *Nr4a1*, a key transcription factor in exhausted T cells dysfunction[64,65], was overexpressed in AhR-deficient 'progenitor exhausted' T cells (Fig. 6d). To validate the impact of diet-derived AhR ligands on T cells respiration functions, we analyzed their ATP content upon anti-PD1 treatment. While mitochondria mass was similar between diet groups and irrespective of treatment (Fig. 6e), 'progenitor exhausted' CD8 T cells displayed a significant increase in ATP content upon anti-PD1 treatment, only in mice fed on I3C diet (Fig. 6f). We also assessed the in vivo proliferation of tumor-infiltrating CD8 T cells from mice fed on indole-poor or I3C diet, using Edu incorporation. Proliferation of 'progenitor exhausted' CD8 T cells after anti-PD1 treatment was significantly impaired in mice fed on indole-poor diet (Fig. 6g and Supplementary Fig. 8a). Collectively, these results indicate that physiological activation of AhR by dietary ligands promotes 'progenitor exhausted' T cells reinvigoration.

To address the relevance of these findings for human, we re-analyzed scRNA-seq data of tumor-infiltrating CD8 T cells in clinical samples from non-small-cell lung cancer (NSCLC) patients after treatment with anti-PD1[55]. We used the 236 genes that we found overexpressed in WT 'progenitor exhausted' CD8 T cells as a transcriptomic signature for 'AhR-dependent genes', and assessed its enrichment across human CD8 T cells clusters as defined in the original study (non-exhausted, terminally exhausted and proliferating cells). Our gene signature was enriched in proliferating CD8 T cells, but only in responder patients (Fig. 6f), suggesting that AhR-dependent genes are associated with response to anti-PD1 therapy in patients.

### AhR licenses the functional response of effector CD8 T cells to anti-PD1 treatment

PD1 blockade can also impact PD1⁻ CD8 T cells indirectly, with an important role for these cells in the subsequent immune response[66].

Therefore, we also analyzed PD1⁻ CD62L⁻ CD8 effector T cells from Lck*AhR^Δ mice and WT littermates. AhR deficiency had a pronounced effect on their transcriptome (514 genes enriched in WT cells and 483 genes enriched in KO cells) (Supplementary Data 3 and Supplementary Data 4). For genes overexpressed in WT cells, pathway enrichment analysis showed an overrepresentation of cytokine responses, including IL-2 signaling and interferon responses (Fig. 7a, b). Genes from these pathways were more expressed in WT effector CD8 T cells, including *Il2ra* and *Il2rb* (encoding IL2-receptor) (Fig. 7c), and multiple interferon response genes, including *Ifnar1* (Fig. 7d). Key genes involved in effector functions were also differentially expressed, including co-stimulatory molecules *Icos* and *Tnfrsf18* (encoding GITR), *Cx3cr1*, *Tbx21* (encoding T-bet), *Fasl*, *Ifng* and multiple granzyme genes (Fig. 7d). Finally, chemotaxis-related genes were enriched in WT effector CD8 T cells (Fig. 7a), in particular *Ccl3* and *Ccl4* (Fig. 7d), which were found to attract NK cells into tumors[67,68]. We then sought to validate these observations at the functional level. We confirmed that granzyme B production upon anti-PD1 treatment was impaired in tumor-infiltrating AhR-deficient CD8 T cells (Supplementary Fig. 8b), contrasting with granzyme B production by WT CD8 T cells being similar in both diet groups (Fig. 4c). We reasoned that activation by non-dietary AhR ligands could contribute to some of the observed transcriptomic differences between WT and AhR-deficient effector CD8 T cells, including granzyme B expression. To directly address the contribution of diet-derived AhR ligands to effector functions, we analyzed the expression of co-stimulatory molecules ICOS and GITR by tumor-infiltrating effector CD8 T cells from mice fed on indole-poor or I3C diet, after anti-PD1 treatment (Fig. 7e). Effector CD8 T cells increased their expression of these molecules upon anti-PD1 treatment only in mice fed on I3C diet. In addition, we measured chemokine concentration in tumors (Fig. 7f). Upon anti-PD1 treatment, CCL3 and CCL4 were significantly increased in tumors from mice fed on I3C diet, and this effect was specific as other chemokines remained unaffected (Supplementary Fig. 8c). Collectively, these results suggest that AhR activation by dietary ligands licenses part of the functional response of effector CD8 T cells to anti-PD1 treatment, in particular potentiating co-stimulation.

## Discussion

In this work, we identified a major role for diet-derived AhR agonists in the reinvigoration of the immune response following anti-PD1 treatment. We also uncovered the underlying mechanisms, showing that AhR was essential in CD8 T cells, favoring the fitness of 'progenitor exhausted' T cells and the functional response of CD8 effector T cells.

AhR signaling in the gut is essential for intestinal homeostasis and barrier function[28]. Deficiency in gut-derived AhR ligands can lead to intestinal inflammation, loss of barrier integrity and dysbiosis[28,69], but also influences the development of inflammatory diseases at distant sites such as neuro-inflammation[21,22] and skin allergy[24]. Orally-administered AhR ligands are distributed systemically[70], including in tumors as reported here, and can exert their effect locally in each

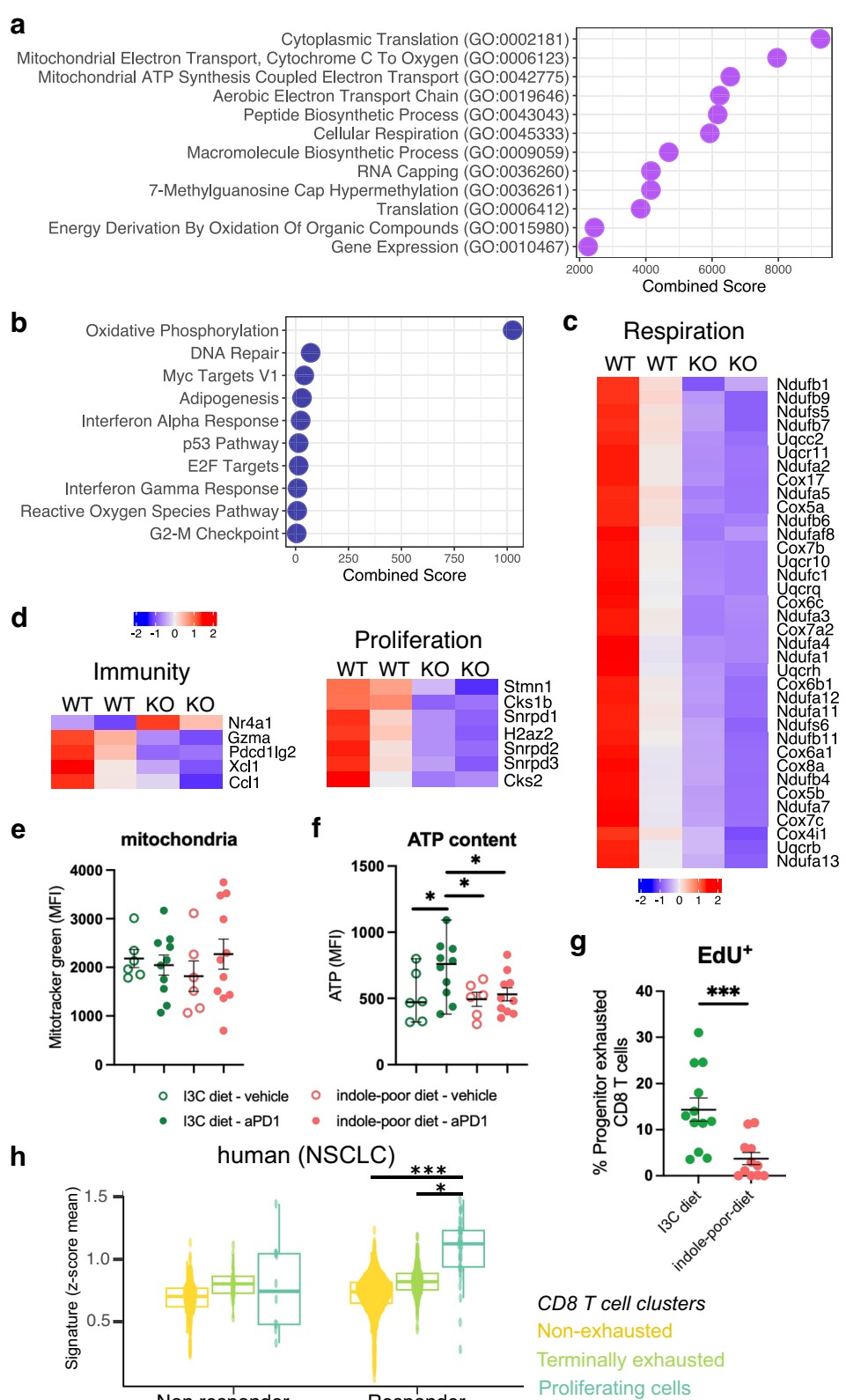

tissue. However, a disrupted intestinal environment might also account for some systemic effects of AhR ligands deficiency, via bacterial translocation or outgrowth and changes in microbiota metabolism. This is the case in metabolic syndrome, where defective AhR ligand production in the gut leads to systemic inflammation and dysfunctional glucose metabolism via translocation of microbial products and reduced hormone production by intestinal epithelial cells[71]. In our experimental setting, we reduced the supply of food-derived AhR agonists, but relative abundance of microbiota-derived AhR ligands from indole and tryptophan metabolism remained unchanged, indicating a selective rather than global deficiency for AhR activation. Consistent with this, intestinal barrier integrity was not affected by the indole-poor diet. In addition, specific deficiency for AhR signaling in T cells recapitulated the phenotype observed with indole-poor diet.

**Fig. 6 | AhR promotes the reinvigoration of 'progenitor exhausted' T cells upon anti-PD1 treatment.** Mice were inoculated with MCA101-OVA tumor cells. Mice were treated when tumors reached 80–100 mm³. **a–d** Lck*AhRᐞ mice (KO) and WT littermates were placed on I3C diet for 3 weeks of adaptation prior to the start of experiments. Tumor-infiltrating 'progenitor exhausted' CD8 T cells were isolated 24 h after anti-PD1 injection and analyzed by RNA-seq (n = 2 independent samples). **a**, **b** Pathways enrichment analysis for genes overexpressed in WT cells using Gene Ontology Biological Processes (**a**) or MSigDB Hallmark (**b**) database. **c**, **d** Scaled expression of selected genes. **e–g** Mice were placed on Indole-poor diet or I3C diet for 3 weeks of adaptation. Tumor-infiltrating cells were analyzed by flow cytometry 24 h (**e**, **f**) or 48 h (**g**) after anti-PD1 injection. Mean fluorescence intensity (MFI) for mitochondria mass (**e**) or ATP (**f**) staining, gated on 'progenitor exhausted' CD8 T cells. Mean +/− sem are shown (n = 6–10 biological replicates in 3 cohorts). One-

way ANOVA. **g** Edu was injected 16 h prior analysis. Percentage of 'progenitor exhausted' CD8 T cells showing Edu incorporation. Mean +/− sem are shown (n = 10–12 biological replicates in 2 cohorts). Two-tailed Mann–Whitney test. **h** ScRNA-seq data of tumor-infiltrating CD8 T cells from non-small-cell lung cancer (NSCLC) patients was reanalyzed for samples obtained post anti-PD1 treatment (GSE179994). Patients were divided into responders (n = 6 patients) and non-responders (n = 5 patients). Signature score across CD8 T cell clusters of the genes found overexpressed in mouse WT 'progenitor exhausted' CD8 T cells. Mean +/− sem are shown, box limits show first quartile and third quartile. Minimum and maximum are indicated by the whiskers. One-way ANOVA with Tukey's correction. For all panels, *p < 0.05, ***p < 0.001. Absence of star indicates 'not significant'. Source data are provided as a Source Data file.

These observations support the idea that dietary AhR ligands impact anti-tumor immune responses in a direct manner, rather than indirectly through alterations of the intestinal mucosa environment.

Our results add to the emerging evidence for a dual role of physiological AhR activation in anti-tumor immune responses. In a melanoma model, indole-3-aldehyde produced by intra-tumoral *Lactobacillus reuteri* promoted anti-PDL1 treatment[35]. This effect was due to increased abundance of IFNγ producing CD8 T cells in the tumor. Indole-3-aldehyde abundance in the serum of melanoma patients also correlated with response to anti-PD1 therapy[35]. By contrast, in a pancreatic cancer model, AhR activation in macrophages by microbiota-derived indoles (indole-3-acetic acid and indole-3-lactic acid) was deleterious for the anti-tumor immune response, while indole-3-aldehyde had no impact[31]. Deletion of AhR in myeloid cells promoted CD8 T cell effector function, and AhR activity correlated with outcome in patients with pancreatic ductal adenocarcinoma[31]. Here we found that lack of dietary AhR ligands was detrimental for the efficacy of anti-PD1 therapy in pre-clinical models of fibrosarcoma, melanoma and mammary tumor. We showed that AhR in T cells, but not in macrophages, was involved. Collectively, these findings suggest that the role of AhR in anti-tumor immunity could be cancer type-specific.

In contrast to the aforementioned studies, we found that imbalance in microbiota-derived AhR ligands did not have a detectable impact on anti-PD1 therapy, while ligands generated from food-derived I3C were required. Distinct roles for dietary and microbiota-derived AhR ligands have been evidenced in other contexts. In keratinocytes, AhR agonists from skin microbiota drive the expression of barrier function molecules[72], while diet-derived I3C controls inflammatory networks[24]. In the gut, lack of dietary I3C decreases intestinal type 3 innate lymphoid cells (ILC) and lymphoid follicles[73], but microbiota-derived AhR ligands are dispensable as these cells are normal in germ-free mice[74]. These observations may be explained by ligand-specific effects or distinct affinity for AhR[75]. More work is needed to disentangle the role of food-derived versus microbiota-derived AhR ligands in immune and non-immune cells.

Our work highlights the importance of a diet with adequate content in indole phytonutrients for favoring the response to immune checkpoint blockade therapy. Our results could also have implications for patients treated with CAR-T cells therapies, as the metabolic fitness of CAR-T cells is a critical parameter for the efficacy of this therapy[76]. Beyond cancer, PD1 blockade has shown therapeutic potential for chronic infections (such as hepatitis B, HIV and blood-stage malaria) and sepsis[77]. Our preclinical findings will pave the way for optimizing the diet of patients treated with immunotherapies to increase efficacy and proportion of responders.

## Methods
### Study design
Sample-size calculation was performed using InVivo stat software (v4.2). Experiments were replicated independently. No data was

excluded. No randomization method was used. The number of biological replicates and of independent experiments is indicated in each legend. For all experiments in this manuscript, samples or animals were allocated to experiments and experimental groups randomly. For all experiments, investigators were blinded during data collection by giving each sample analyzed an arbitrary number. Blinding was not possible for analyzing tumor growth kinetics, because measurements were performed longitudinally for each mouse.

### Mouse strains
C57BL/6J mice (strain 000664) were obtained from Charles River (France). *Lysm*-Cre (strain 004781), *Lck*-Cre (strain 003802) and *Ahr*-flox (strain 006203) mice were obtained from Jackson Laboratory. *Ncr1*-iCre mice[78] and Ncr1*Il2rgᐞ mice[49] were maintained in-house at Institut Pasteur. Lysm*AhR, Lck*AhR and Ncr1*AhR strains were generated in-house by crossing *Ahr*-flox with *Lysm*-Cre *Lck*-Cre or *Ncr1*-iCre mice, respectively. AhR-Tomato reporter mice (strain BSEC AHR Tomato) were obtained from B.Stockinger (King's College London) and have been described previously[50]. All mice were on C57BL/6 background. Mice were maintained under specific pathogen-free conditions at the animal facility of Institut Curie or Institut Pasteur in accordance with institutional guidelines. Animal care and use for this study were performed in accordance with the recommendations of the European Community (2010/63/UE) for the care and use of laboratory animals. Mice were housed in a 12 h light/12 h dark environment at a temperature of 22 °C (+/−2 °C) and humidity of 55% (+/−10%). Only female mice were used and euthanized by cervical dislocation at age 8–12 weeks. All animal procedures were in accordance with the guidelines and regulations of the French Veterinary Department and have been approved by the local ethics committee (CEEA118 Comité Recherche et Ethique Animale de l'institut Curie, authorization APA-FiS#14331-20180330114641-v2).

Mice were maintained on a purified diet ('indole-poor diet', AIN-93M, U8978 Version 55, Safe diets), or the same diet supplemented at 200 ppm in Indole-3-carbinol (I3C, # I7256, Sigma) termed 'I3C diet'. Mice were placed on the specific diet at 5 weeks of age for a period of adaptation of 3 weeks, before being used for any experiment. In some experiments, mice were fed on a standard chow diet (#4RF25SV-PF1609, aliment pellets, Mucedola). Mice had *ad libitum* access to water and food. In some experiments, mice were fed on 'indole-poor diet' and supplemented by oral gavage with I3C, 3,3'-Diindolylmethane (DIM, #D9568) or Indole-3-acetic acid (IAA, # I3750) (all from Sigma) 3 times a week (200 µg/mouse, diluted in corn oil, #C8267, Sigma), or the same volume of corn oil as control. In some experiments, mice were treated with 1 g/L of veterinary Ampicillin (Ampisol, Dopharma France) diluted in drinking water with added sucrose (2 g/L). Ampicillin was administered continuously, starting 5 days before tumor inoculation. Sucrose (2 g/L) was added to the drinking water of untreated mice.

For in vivo NK cell depletion, mice were treated intra-peritoneally with 120 µg of anti-mouse NK1.1 (clone PK136, #BE0036) or of

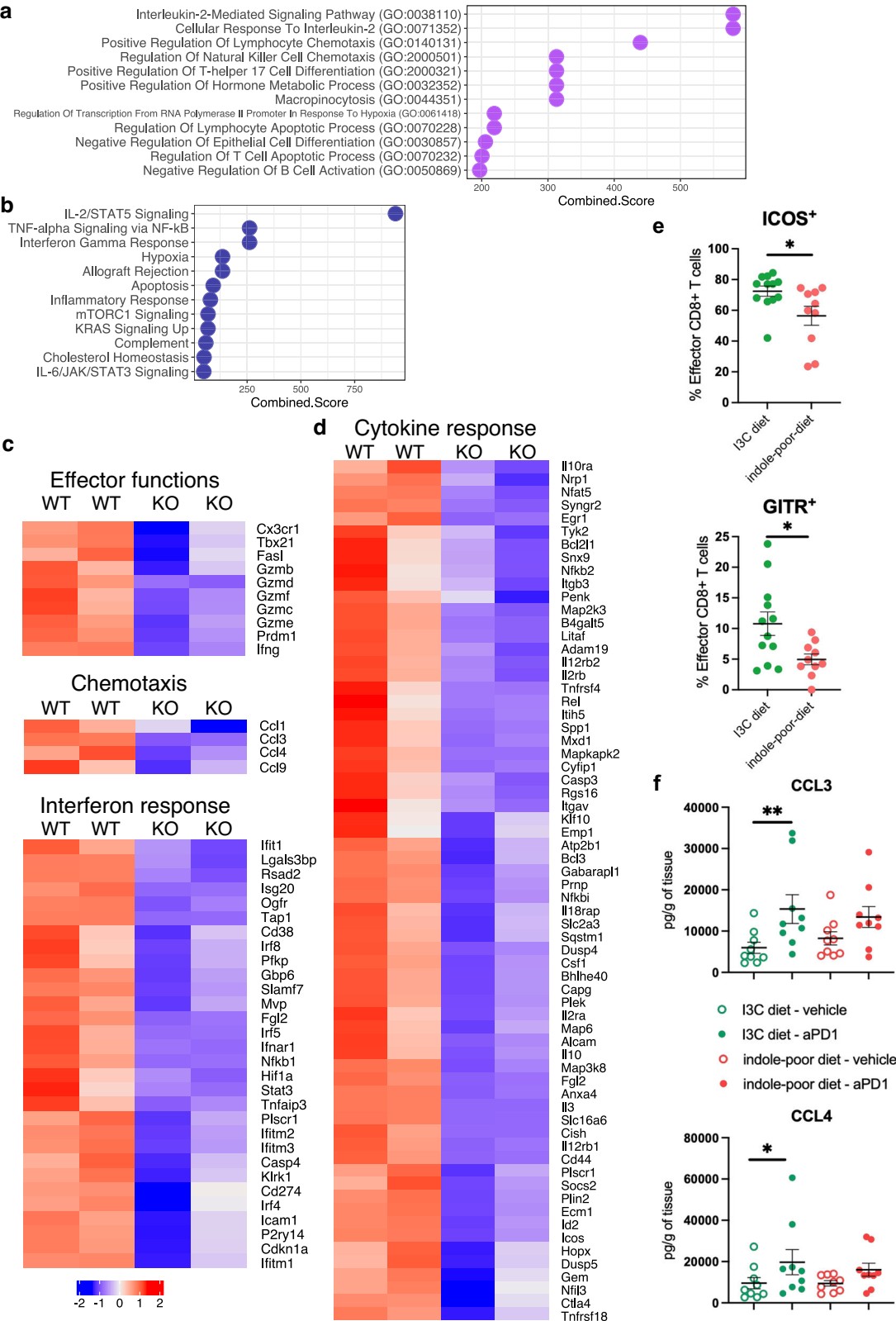

isotype control antibody (clone C1.18.4, #BE0085) (both from BioXcell) diluted in Phosphate Buffer Saline (PBS, #10010015, Gibco). Mice were treated 6 times, on the day before anti-PD1 injection, then on the same days as anti-PD1 injection, and 2 and 4 days after the last treatment. Efficient NK cell depletion was confirmed by flow cytometry staining on blood.

For all animal experiments, the experimental unit is a single animal.

### Reconstitution of germ-free mice

Wild-type *E. coli* BW25113 strain and the single-gene knockout JW3686 mutant invalidated for the *tnaA* gene[79] were provided by the National Institute of Genetics (Mishima, Japan). Strains were stored at −80 °C in Brain Heart Infusion broth (Sigma) with added glycerol (final concentration 40%). For inoculation of germ-free mice, bacteria were grown in lysogeny broth overnight at 37 °C in aerobic conditions.

**Fig. 7 | AhR regulates the functional response of effector CD8 T cells to anti-PD1 treatment.** Mice were inoculated with MCA101-OVA tumor cells. Mice were treated when tumors reached 80–100 mm³. **a**–**d** Lck*AhR$^\Delta$ mice (KO) and WT littermates were placed on I3C diet for 3 weeks of adaptation prior to the start of experiments. Tumor-infiltrating PD1⁻ CD62L⁻ effector CD8 T cells were isolated 24 h after anti-PD1 injection and analyzed by RNA-seq (*n* = 2 independent samples). Pathways enrichment analysis for genes overexpressed in WT cells using Gene Ontology Biological Processes (**a**) or MSigDB Hallmark (**b**) database. **c**, **d** Scaled expression of selected genes. **e**, **f** Mice were placed on Indole-poor diet or I3C diet for 3 weeks of adaptation. **e** Tumor-infiltrating cells were analyzed by flow cytometry 48 h after anti-PD1 injection. Percentage of effector CD8 T cells showing ICOS or GITR staining. Mean +/− sem are shown (*n* = 10–12 biological replicates in 2 cohorts). Two-tailed Mann–Whitney test. **f** Tumors were lysed 48 h after anti-PD1 injection. Chemokine concentration was assessed by cytometric bead array. Mean +/− sem are shown (*n* = 9 biological replicates in 3 cohorts). Kruskall–Wallis test. For all panels, *$p < 0.05$, **$p < 0.01$. Absence of star indicates 'not significant'. Source data are provided as a Source Data file.

Bacterial cultures were distributed in sterile vials and transferred in sterile conditions into the isolators. Germ-free mice were produced in-house at the ANAXEM facility (Micalis Institute, INRAE, Jouy-en-Josas, France). At weaning, germ-free mice were inoculated orally with the wild-type or mutant strain culture using a flexible feeding needle (200 µL/mouse). Gut colonization was checked 10 days later by microscopic examination and cultures of freshly voided feces. Procedures were in accordance with the guidelines and regulations of the French Veterinary Department and have been approved by the local ethics committee (comité d'éthique du center de recherches de Jouy-en-Josas COMETHEA, authorization APAFiS#25483-20200409111941 12-v2).

## Tumor growth analysis
Mice were inoculated intra-dermally at day 0 with $5 \times 10^5$ MCA101-OVA fibrosarcoma cells[80], or with E0771 mammary (ATCC CRL-3461) or with B16F10 (ATCC CRL-6475) tumor cells. All tumor cell lines were established from female mice. When tumors reached 80–100 mm³ (day 7 or 8 depending on experiments), mice were treated intra-peritoneally with 200 µg anti-PD1 (#BE0146, BioXcell) or 100 µL vehicle (PBS), using 3 injections 3 days apart. In rechallenge experiments, mice did not receive any treatment. Tumor volume was measured 3 times a week using a digital calliper. Tumor volume was calculated using the formula: Volume = (length × width²)/2. Mice were classified in 3 categories: tumor progression, rejection or partial response. Partial response was defined as a continuous decrease in tumor volume for a minimum period of 7 days before resuming progression. Percentage of change in tumor volume was calculated using the formula: (volume at day 10 post first treatment dose − volume at first day of treatment)/ volume at first day of treatment*100. Ethical endpoint was defined as tumor volume above 1500 mm³, or at day 32 after tumor inoculation.

## Flow cytometry
Cells were stained with indicated antibody cocktails (Supplementary Data 5) supplemented with Fc block (#553141, BD Biosciences) in FACS buffer consisting of PBS containing 0.5% Bovine Serum Albumin (#05470, Sigma) and 2 mM EDTA (#E7889, Sigma) for 30–45 min on ice. After washing with FACS buffer, cells were resuspended in FACS buffer containing DAPI (#D1306, Fischer Scientific, 100 ng/mL), unless otherwise indicated. Cells were acquired on a ZE5 instrument (Bio-Rad), Aurora (Cytek) or FACSVerse (BD Bioscience). Supervised analysis was performed using FlowJo software v10 (FlowJo LLC).

## Tumor cells isolation
Mice were inoculated subcutaneously with $5 \times 10^5$ MCA101-OVA fibrosarcoma cells at day 0. In some experiments, mice were also inoculated with MCA101 cells containing a mock plasmid. When tumors reached 80–100 mm³ (day 7 or 8 depending on experiments), mice were treated intra-peritoneally with 200 µg anti-PD1 or 100 µL vehicle (PBS). In some experiments, tumors from untreated mice were analyzed. Tumors were cut in small pieces with scalpels and digested for 30 min at 37 °C in digestion mix: RPMI containing 0.4 mg/mL DNAse I (#11284932001, Sigma) and 0.5 mg/mL collagenase

D (#11088858001, Roche). Cell suspensions were then filtered using 40µm cell strainers.

## Tumor-infiltrating cells characterization
For phenotyping, cells were stained with a myeloid panel or a lymphoid panel. After surface staining, cells were washed, stained with fixable Live/Dead NIR dye (#L10119, Thermo Fisher Scientific) for 10 min at 4 °C, fixed and permeabilized (Intracellular Fixation and Permeabilization Buffer Set, #88-8824-00, eBioscience). Cells were then stained for intracellular markers in a buffer containing 2% normal mouse serum (#24-5544-94, Thermo fisher Scientific) and mouse Fc block (#553141, BD Biosciences) for 90 min at room temperature. Myeloid panel: anti-CD26 BV605 (BD Biosciences, clone H194-112), anti-Ly6G BV510 (Biolegend, clone 1A8), anti-CD11c BV786 (BD Biosciences, clone HL3), anti-Ly6C Alexa700 (Biolegend, clone HK1.4), anti-MHC II Pe-Cy5 (Biolegend, clone M5/114.15.2), anti-CD64 PerCP-eFluor710 (eBioscience, clone X54-5/7.1), anti-CD206 BV650 (Biolegend, clone C068C2), anti-Arg1 APC (eBioscience, clone A1exF5), anti-CD11b BV750 (BD Bioscience, clone M1/70), anti-CD45 AF532 (eBioscience, clone 30-F11). Lymphoid panel: anti-NK1.1 BV605 (BD Bioscience, clone PK136), anti-TCRβ FITC (BD Bioscience, clone H57-597), anti-CD8 BV711 (BD Bioscience, clone 53-6.7), anti-CD4 BV650 (Biolegend, clone GK1.5), anti-CD103 PerCP-Cy5.5 (Biolegend, clone 2E7), anti-Foxp3 APC (eBioscience, clone FJK-16s), anti-Lag3 PE-Dazzle594 (Biolegend, clone C9B7W), anti-PD1 BV421 (Biolegend, clone 29F.1A12), anti-CD45 AF532 (eBioscience, clone 30-F11). Samples were acquired on a Aurora instrument (Cytek).

For tetramer staining, CD45+ cells were enriched using magnetic selection (#130-118-780, CD45 TIL, Miltenyi Biotech). Cells were stained with H-2 K$^b$ SIINFEKL Tetramer PE (#TS-5001-1C, MBL international) for 30 min at room temperature, then Fc block (BD Bioscience) was added and cells were stained with anti-CD45 V500 (BD Bioscience, clone 30-F11), anti-CD8 APC-Cy7 (BD Pharmingen, clone 53-6-7), anti-TCRβ FITC (BD Bioscience, clone H57-597), anti-CD4 PerCP-Cy5.5 (Biolegend, clone GK1.5). Samples were acquired on a FACSVerse instrument (BD Bioscience).

For cytokine intracellular staining, cells were either analyzed directly after isolation or after culture (for assessing IFNγ in T cells) in X-VIVO15 serum-free medium (#BEBP02-054Q, Lonza) for 3 h in presence of Phorbol 12-myristate 13-acetate (50 ng/mL, #P8139), ionomycin (1 µg/mL, #407950) and Brefeldin A (4 µg/mL, #203729) (all from Sigma). Cells were first stained with anti-CD8 BUV 395 (BD Bioscience, clone 53-6.7), anti-TCRβ BUV737 (BD Bioscience, clone H57-597), anti-CD45 FITC (BD Biosciences, clone 30-F11), anti-NKp46 BV605 (Biolegend, clone 29A1.4), anti-CD11b PE-CF594 (BD Bioscience, clone M1/70), anti-CD49b APC (BD Bioscience, clone DX5). Cells were then washed, stained with fixable Live/Dead NIR dye (Thermo Fisher Scientific) for 10 min at 4 °C, fixed and permeabilized (Intracellular Fixation and Permeabilization Buffer Set, eBioscience). Cells were then stained for intracellular cytokines in a buffer containing 2% normal mouse serum (Thermo fisher Scientific) and mouse Fc block (BD Biosciences) for 90 min at room temperature with the following antibodies: anti-Granzyme B BV421 (BD Bioscience, clone GB11), anti-IFNγ BV785 (Biolegend, clone XMG1.2). Cells treated according to the same procedure but incubated without the intracellular staining antibodies

were used as background control. Samples were acquired on a ZE5 instrument (Biorad) using volumetric counting.

For the analysis of AhR-TdTomato mice, cells were stained with anti-TCRβ BUV737 (BD Bioscience, clone H57-597), anti-CD8 BUV 395 (BD Bioscience, clone 53-6.7), anti-NK1.1 BV480 (BD Bioscience, clone PK136), anti-CD11b PerCP-Cy5.5 (BD Bioscience, clone M1/70), anti-Ly6G BV605 (Biolegend, clone 1A8), anti-CD11c BV786 (BD Biosciences, clone HL3), anti-CD45 FITC (BD Biosciences, clone 30-F11), anti-CD26 PE (Biolegend, clone DPP-4), anti-Ly6C Alexa700 (Biolegend, clone HK1.4), anti-MHC II APC-Cy7 (Biolegend, clone M5/114.15.2), anti-CD64 Pe-Cy7 (eBioscience, clone X54-5/7.1), anti-CD49b APC (BD Bioscience, clone DX5). Samples were acquired on a ZE5 instrument (Biorad).

For the analysis of 'progenitor exhausted' and 'effector' CD8+ T cells, cells were stained for extracellular markers in a buffer containing PBS, 2% fetal calf serum (#CA-115, Biosera) and 2 mM EDTA. After surface staining, cells were washed, stained with fixable Live/Dead Zombie NIR dye (#L10119, Thermo Fisher Scientific) for 10 min at 4 °C, and fixed in PBS 4% paraformaldehyde (diluted from 16% paraformaldehyde, #28906, Pierce) for 15 min at 4 °C. Cells were then permeabilized and stained for 5-ethynyl 2'-deoxyuridine (EdU) detection with Alexa fluor 488, using Click-iT Edu cell proliferation assay (#C10337, ThermoFisher) according to manufacturer's instructions. Cells were stained with anti-TCRβ BUV395 (BD Bioscience, clone H57-597), anti-CD11b BV750 (Biolegend, clone M1/70), anti-PD1 BV421 (Biolegend, clone 29F.1A12), anti-CD278/ICOS BUV737 (BD Bioscience, clone C398.4A), anti-CD25 BV605 (Biolegend, clone PC61), anti-CD45 Alexa Fluor 532 (eBioscience, clone 30-F11), anti-CD4 PerCP-Cy5.5 (BD Bioscience, clone RM4-5), anti-CD62L APC (Biolegend, clone MEL-14), anti-CD357/GITR Pe-Cy7 (eBioscience, clone eBioAITR), anti-Slamf6 PE (Biolegend, clone 330-AJ), anti-CD8 APC-Cy7 (Biolegend, clone YTS156.7.7). Effector CD8 T cells were gated as live CD11b− TCRβ+ CD4− CD8+ PD1− CD62L− and 'progenitor exhausted' CD8 T cells were gated as live CD11b− TCRβ+ CD4− CD8+ PD1+ CD62L− Slamf6+. Samples were acquired on an Aurora instrument (Cytek).

For the analysis of mitochondria and ATP content, cells were placed in X-VIVO15 serum-free medium for 30 min at 37 °C in the presence of Mitotracker green (#M7514, Thermo Fisher) or ATP-red (#SCT045, Merck Millipore), respectively. After washing, cells were stained in FACS buffer with anti-CD8 BUV 395 (BD Bioscience, clone 53-6.7), anti-TCRβ BUV737 (BD Bioscience, clone H57-597), anti-PD1 BV421 (Biolegend, clone 29 F.1A12), anti-CD45 PerCPCy5.5 (BD Biosciences, clone 30-F11), anti-Slamf6 PE (Biolegend, clone 330-AJ), anti-CD62L APC (Biolegend, clone MEL-14), anti-CD11b APC-Cy7 (Biolegend, clone M1/70). Samples were acquired on a ZE5 instrument (Biorad).

## Indole measurement in cecal content

Frozen cecal contents of the gnotobiotic rats were thawed, weighed and 10-fold diluted (w/v) in 50 mM PBS. After homogenization, the cecal suspension was centrifuged ($8000 \times g$, 10 min, 4 °C) and the supernatant collected. The pellet was resuspended and centrifuged again twice, and the 3 supernatants were pooled for injection onto the HPLC column (sample volume: 50 μL). Samples were analyzed by HPLC using a 2690 autosampler and separation module equipped with a 474 fluorescence detector and a 996 photodiode array detector (Waters). Separation of indole was carried out in a reversed-phase column packed with LiChrospher R 100RP-18e (5 μm, 250 × 4.3 mm; VWR), equipped with a guard column LiChrospher R 100 RP-18e (5 μm, 4 × 4 mm; VWR). Elution was isocratic (55% methanol and 45% ultrapure water) at a flow rate of 0.5 mL/min. The excitation and emission wavelengths of the detector were set at 285 and 320 nm, respectively. Indole eluted at 16.7 min and was quantified using external standard curves. Results were expressed as nmol/g of dry matter (DM). DM was determined by weighing cecal content

samples before and after 24-h freeze-drying (VirTis Lyophilizer, SP Scientific).

## Indole measurement in tumors and blood

Blood samples were centrifuged for 15 min at $1200 \times g$ to obtain serum. Tumors were collected when they reached 80–100 mm³ and snap frozen. 50 mg of frozen tumor were placed in a 2 mL Eppendorf tube with 500 μL of cold ammonium bicarbonate. One homogenization bead (tungsten carbide bead, 3 mm, #69997 Qiagen) was added before homogenizing in a Tissuelyser II (#9003240, Qiagen) at 30 Hz for 2 min. Homogenate was transferred into a new tube. Leftover material was washed with another 500 μL of ammonium bicarbonate and pooled with the homogenate. Protein content was measured using Pierce BCA Protein Assay Kit (#23225, Thermofisher).

Indoles levels were measured in samples using liquid chromatography coupled to high-resolution mass spectrometry (HPLC-HRMS)[81]. Briefly, 2 mg of protein were dispersed in 450 μL of water and added to 450 μL of methanol and samples were vortexed for 5 min in a thermomixer at 4 °C followed by an incubation at −20 °C for 30 min. After centrifugation at $13,300 \times g$ for 10 min, the supernatant was collected and concentrated using a SpeedVac vacuum concentrator (Thermo Scientific). Samples were resuspended in 100 μL of 10% methanol solution, vortexed in a thermomixer for 5 min and centrifuged at $13,300 \times g$ for 10 min. 80 μL of each sample was transferred to a liquid chromatography vial and 10 μL injected in the HPLC-HRMS system. Chromatography was carried out with a Phenomenex Kinetex 1.7 μm XB – C18 (150 mm × 2.10 mm) and 100 Å HPLC column maintained at 55 °C. The solvent system comprised mobile phase A [0.5% (vol/vol) formic acid in water], and mobile phase B [0.5% (vol/vol) formic acid in methanol]. The gradient was set-up as follows: 0–2 min, 0% B; 2–7 min, 0–50% B; 7–15 min, 50–100% B; 15–18 min, 100% B, 18–18.5 min 100–0% B and 18.5–21.5 min 0%B. HRMS analyses were performed on a HPLC Vantage Flex (Thermo Fischer Scientific) coupled to a Q-Exactive Focus mass spectrometer (Thermo Fisher Scientific) that was operated in positive (ESI+). The HPLC autosampler temperature was set at 4 °C. The heated electrospray ionization source was set with a spray voltage of 4.5 kV, a capillary temperature of 250 °C, a heater temperature of 475 °C, a sheath gas flow of 35 arbitrary units (AU), an auxiliary gas flow of 10 AU, a spare gas flow of 1 AU, and a tube lens voltage of 100 V. During the HRMS acquisition, the scan range was set to m/z = 100–500 Da, the instrument operated at 70,000 resolution (m/z = 200), with an automatic gain control (AGC) target of $1 \times 10^6$ and a maximum injection time (IT) set to automatic. Instrumental chromatography stability was evaluated by injection of a synthetic standard mixture with all metabolites of interest and quality control (QC) samples quality control obtained from a pool of the left-over of all samples analyzed. This QC sample was reinjected once at the beginning of the analysis, every 10 sample injections, and at the end of the run. Ionization and retention times were validated with pure standards (Table 1).

## Chemokine analysis in tumors

Tumors were placed in a 2 mL Eppendorf tube with 500 μL of Assay buffer from CBA kit (#558266, BD Bioscience). One homogenization bead (tungsten carbide bead, 3 mm, Qiagen) was added before homogenizing in a Tissuelyser II (Qiagen) at 30 Hz for 2 min. Homogenate was transferred into a new tube. Leftover material was washed with another 500 μL of assay buffer and pooled with the homogenate. Tumor homogenates were filtered before storing at −20 °C. CCL3/MIP-1α (#558449), CCL4/MIP-1β (#558343), CXCL9/MIG (#558341) and CXCL1/KC (#558340) concentration was measured using CBA assay (all from BD Biosciences). The limit of detection was 2.3 pg/mL.

## Intestinal permeability assay

Mice were starved for 12 h then gavaged with 200 μL of 100 mg/mL 4 kDa FITC Dextran (#53471, Sigma) diluted in water. After 2 h, blood

**Table 1 | Parameters used for validation for each metabolite**

| Metabolite | m/z | Adduct | Polarity | Retention time (min) |
|---|---|---|---|---|
| Indole-3-carbinol | 130.06513 | Water loss | pos | 9.7 |
| Indol-3-carboxaldehyde | 146.06004 | +H$^+$ | pos | 8.7 |
| L-Kynurenine | 209.09207 | +H$^+$ | pos | 4.1 |
| 3,3′ –Diindolyl-methane | 130.06513 | In-source fragment | pos | 12.5 |
| Indole-3-acetic acid | 176.07061 | +H$^+$ | pos | 8.9 |
| Indole-3-propionic acid | 190.08626 | +H$^+$ | pos | 9.7 |
| Indole-3-lactic acid | 206.08117 | +H$^+$ | pos | 8.6 |
| Indoxyl-sulfate | 212.0023 | –H$^+$ | neg | 8.0 |

Peak area was used as read out for relative quantification of the metabolites across the samples and normalized by total protein amount as assessed by BCA assay.

was collected in heparin-coated tubes. Plasma was obtained by centrifugation (2000 × $g$, 5 min). FITC fluorescence in plasma was measured using a spectrophotometer (excitation 485 nm, emission 528 nm). Concentration was determined against a standard curve.

### Isolation of intestinal cells
Duodenum or colon samples (2.5 cm in length) were cut in small pieces with scalpels and digested for 30 min at 37 °C in digestion mix: RPMI containing 0.4 mg/mL DNAse I (#11284932001, Sigma) and 0.5 mg/mL collagenase D (#11088858001, Roche). Cell suspensions were then filtered using 40 μm cell strainers (#CLS352340, Falcon).

### Lymph node cells analysis
Tumor-draining lymph nodes were cut into small pieces and digested for 30 min at 37 °C in digestion mix: RPMI containing 0.4 mg/mL DNAse I (#11284932001, Sigma) and 0.5 mg/mL collagenase D (#11088858001, Roche). Cell suspensions were then filtered using 40 μm cell strainers. Cells were stained with H-2 K$^b$ SIINFEKL Tetramer PE (#TS-5001-1C, MBL international) for 30 min at room temperature, then Fc block (#553141, BD Bioscience) was added. For analysis prior to therapy, cells were stained with anti-TCRβ PeCy7 (BD Bioscience, clone H57-597), anti-CD8 APC (BD Pharmingen, clone 53-6-7), anti-CD11c FITC (BD Pharmingen, clone HL3), anti-XCR1 PerCP-Cy5.5 (Biolegend, clone ZET), anti-CD103 BV510 (Biolegend, clone 2E7), anti-CD11b PE-Texas Red (BD Bioscience, clone M1/70), anti-MHC II APC Cy7 (Biolegend, clone M5/114.15.2). Samples were acquired on a FACSVerse instrument (BD Bioscience). For analysis after therapy, cells were stained with anti-TCRβ BUV737 (BD Bioscience, clone H57-597), anti-CD8 BUV 395 (BD Bioscience, clone 53-6-7), anti-CD19 BV480 (BD Bioscience, clone 1D3), anti-CD11b PerCP-Cy5.5 (BD Bioscience, clone M1/70). Samples were acquired on a ZE5 instrument (Biorad).

For the analysis of AhR-TdTomato mice, cells were stained with anti-TCRβ BUV737 (BD Bioscience, clone H57-597), anti-CD8 BUV395 (BD Bioscience, clone 53-6-7), anti-CD19 BV480 (BD Bioscience, clone 1D3), anti-CD4 PerCP-Cy5.5 (BD Bioscience, clone RM4-5), anti-CD11b Pe-Cy7 (BD Bioscience, clone M1/70). Samples were acquired on a ZE5 instrument (Biorad).

### Bone marrow cells analysis
For the analysis of AhR-TdTomato mice, cells were stained with anti-TCRβ BUV737 (BD Bioscience, clone H57-597), anti-CD8 BUV395 (BD Bioscience, clone 53-6-7), anti-NK1.1 BV480 (BD Bioscience, clone PK136), anti-CD11b Pe-Cy7 (BD Bioscience, clone M1/70), anti-Ly6G BV605 (Biolegend, clone 1A8), anti-Ly6C FITC (Biolegend, clone HK1.4), anti-CD115 APC (BD Bioscience, clone AFS98), anti-CD43 BB700 (BD Bioscience, clone S7). Samples were acquired on a ZE5 instrument (Biorad).

### Spleen NK cell analysis
For flow cytometry analysis, cells were stained for viability with Fixable Viability Dye eFluor 506 (#65-0866-14, eBioscience) in PBS during

15 min on ice. Cells were washed with FACS buffer, then stained for extracellular markers in the presence of anti-FcR (#BE0307, anti-CD16/32, clone 2.4G2, BioXcell) for 45 min on ice. The following antibodies were used: anti-CD3 BV605 (BD Biosciences, clone 145-2C11), anti-CD5 APC-R700 (BD Biosciences, clone 53-7.3), anti-CD11b FITC (eBioscience, clone M1/70), anti-CD19 BV786 (BD Biosciences, clone 1D3), anti-CD27 V450 (BD Biosciences, clone LG3.A10), anti-CD45 BUV395 (BD Biosciences, clone 30F10), anti-Ly49H Pe-CF594 (BD Biosciences, clone 3D11), anti-Ly49D APC (eBioscience, clone 4E5), anti-NKp46 PerCP-eFluor710 (eBioscience, clone 29A1.4), anti-Ly49G2 Pe-Cy7 (BD Biosciences, clone 4D11), anti-Ly49C/I PE (BD Biosciences, clone 5E6).

For ex vivo stimulation, 10$^6$ freshly isolated spleen cells were cultured in round-bottom microtiter plates and stimulated for 4 h at 37 °C in RPMI 10% FCS. For detection of IFNγ, cells were stimulated with IL-12 (1 μg/mL, #10018-IL, R&D systems) and IL-18 (10 μg/mL, #9139-IL, R&D systems) alone or in combination (IL-12 + IL-18), or with PMA (#P1585, Sigma) and Ionomycin (#407950, Sigma). For detection of CD107a, monensin (#554724, GolgiStop, BD Biosciences) together with brefeldin A (#555029, GolgiPlug, BD Biosciences) were added together with anti-CD107a FITC (BD Biosciences, clone: 1D4B) at the beginning of the assays. At the end of stimulation, cells were washed and stained for extracellular markers followed by fixation using BD Cytofix/Cytoperm (#554714, BD Biosciences), and finally stained intracellularly with anti-IFNγ Pe-Cy7 (BD Biosciences, clone: XMG1.2) diluted in BD Cytoperm.

### qPCR
For qPCR analysis of MCA101-OVA tumor cells, cells were exposed for 8 h to 8 μM SR1 (#10625, Cayman chemicals) or 60 nM FICZ (#BML-GR206, Enzo Life Sciences) or 50 μM DIM (#D9568, Sigma) or the same volume of DMSO (vehicle). For comparison, 5 × 10$^5$ monocytes were exposed for 8 h to 60 nM FICZ, after purification from C57BL/6 bone marrow with EasySep mouse monocyte isolation kit (#19861, Stemcell).

Cells were lysed in RLT buffer (#79216, QIAGEN). RNA extraction was carried out using the RNAeasy micro kit (#74004, QIAGEN) according to manufacturer's instructions. Total RNA was retro-transcribed using the superscript II polymerase (#18064071, Invitrogen), in combination with random hexamers, oligo(dT) (#C1181, Promega) and dNTPs (#U1205, #U1225, #U1215 and #U1235, Promega). Transcripts were quantified by real time PCR on a 480 LightCycler instrument (Roche). Reactions were carried out in 10 μL, using a Mastermix (Eurogentec), with the following Taqman Assays primers (#4331182, ThermoFisher): *Gapdh* (Mm99999915_g1), *Polr2a* (Mm00839502_m1), *Ccl2* (Mm00441242_m1), *Mmp12* (Mm00500554_m1), *Gzmb* (Mm0042837), *Nos2* (Mm00440502_m1), *Ifng* (Mm01168134_m1), *Entpd1* (Mm00515447_m1), *Ccl5* (Mm01302427_m1), *Cxcl9* (Mm00434946_m1), *Arg1* (Mm00475988_m1), *Vegf* (Mm00437306_m1), *Cyp1a1* (Mm00487218_m1), *Ahrr* (Mm00477443_m1). The second derivative method was used to determine each Cp and the expression of genes of interest relative to the housekeeping genes

*Gapdh* (Mm99999915_g1) *and Polr2a* (Mm00839502_m1) was quantified.

## Single cell RNA-seq library preparation

Cellular suspension (3000 cells) was loaded on a 10X Chromium instrument (10X Genomics) according to manufacturer's protocol. Single-cell RNA-Seq libraries were prepared using Chromium Single Cell 3' v3.1 Reagent Kit (10X Genomics) according to manufacturer's protocol. Library quantification and quality assessment was performed using Qubit fluorometric assay with dsDNA High Sensitivity Assay Kit (#Q3323, Invitrogen) and Bioanalyzer Agilent 2100 using a High Sensitivity DNA chip (#5067-4626, Agilent Genomics). Indexed libraries were equimolarly pooled and sequenced on an Illumina NovaSeq 6000 using paired-end 26x91bp as sequencing mode.

## Single-cell RNA-seq data analysis

Single-cell expression was analyzed using the Cell Ranger Single Cell Software Suite (v6.0) to perform quality control, sample de-multiplexing, barcode processing, and single-cell 3' gene counting. Further analysis was performed in R (v4.1.0) using the Seurat package (v4.0.3). The gene-cell-barcode matrix of the samples was log-transformed and filtered based on the number of genes detected per cell (any cell with less than 200 genes or more than 2500 genes per cell was filtered out). Regression in gene expression was performed based on the number of unique molecular identifiers (UMI) and the percentage of mitochondrial genes. Only genes detected in at least 3 cells were included. Cells were then scaled to a total of 1e4 molecules. Any cell with more than 5% of mitochondrial UMI counts was filtered out. PCA was run on the normalized gene-barcode matrix. The first 15 principal components were used for the UMAP projection and clustering analysis using the Elbow Plot approach. Clusters were identified using the « FindClusters » function in Seurat with a resolution parameter of 0.8 for both datasets. Unique cluster-specific genes were identified by running the Seurat « FindAllMarkers » function. Differentially expressed genes were identified based on adjusted *p*-value < 0.05 and Log2 FoldChange > 0.25. Heatmaps and plots were plotted using Seurat. Data is available at GEO (GSE271049).

## RNA-seq library preparation

For RNA-seq analysis of T cells, samples were prepared by pooling tumors from 4 individual mice in each group. Cells were stained anti-CD11b BV421 (Biolegend, clone M1/70), anti-CD3 BV510 (Biolegend, clone 17A2), anti-PD1 FITC (Biolegend, clone 29F.1A12), anti-CD4 PerCP-Cy5.5 (BD Bioscience, clone RM4-5), anti-CD62L PeCy7 (Biolegend, clone MEL-14), anti-Slamf6 PE (Biolegend, clone 330-AJ), anti-CD8 APC-Cy7 (Biolegend, clone YTS156.7.7). DAPI was used to exclude dead cells. Effector CD8 T cells were gated as live CD11b− CD3+ CD4− CD8+ PD1− CD62L− and 'progenitor exhausted' CD8 T cells were gated as live CD11b− CD3+ CD4− CD8+ PD1+ CD62L− Slamf6+. Cells were sorted on a FACSAria instrument (BD Bioscience). Total RNA was extracted using the Single Cell RNA Purification Kit (#SKU51800, Norgen) according to the manufacturer's protocol. cDNA was generated using the SMART-Seq v4 kit (#634890, Takara) according to the manufacturer's protocol. The integrity of the DNA was confirmed in BioAnalyzer using High Sensitivity DNA kit (#5067-4626, Agilent Technologies) (8 < RIN). Libraries were prepared according to Illumina's instructions accompanying the Nextera XT DNA Library Prep Kit (Illumina). Sequencing was performed in 2 sequencing unit of NovaSeq 6000 (Illumina) (100-nt-length reads, paired end).

## RNA-seq data analysis

Genome assembly was based on the Genome Reference Consortium (mm10). Quality of RNA-seq data was assessed using FastQC (v0.11.9). Reads were aligned to the transcriptome using STAR[82]. Differential gene expression analysis was performed using DESeq2 (v1.34)[83]. Genes with low number of counts (<5) were filtered out. Differentially expressed genes were identified based on adjusted *p*-value < 0.01 and Log2 FoldChange >0.5. Heatmaps of log2-scaled expression were generated with ComplexHeatmap. Pathway enrichment was analyzed using EnrichR[84]. Data is available at GEO (GSE271049).

## Statistical analysis

Statistical tests were performed using Prism v10 (GraphPad Software). Statistical details for each experiment can be found in the corresponding figure legend. Two-sided testing was used. Specifically, two-tailed Mann–Whitney test, Kruskall–Wallis test, or one-way ANOVA is used wherever appropriate. N corresponds to the number of biological replicates.

## Reporting summary

Further information on research design is available in the Nature Portfolio Reporting Summary linked to this article.

## Data availability

All data are included in the Supplementary Information or available from the authors, as are unique reagents used in this Article. The raw numbers for charts and graphs are available in the Source Data file whenever possible. Sequencing data have been deposited in GEO under the accession codes GSE271049 for RNA-seq and GSE271049 for scRNA-seq. Other data that support the findings of this study are available from the corresponding author upon reasonable request. Source data are provided with this paper.

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

## Acknowledgements

The authors wish to thank the Flow Cytometry Core, the NGS Platform and the In vivo Experiments Platform of Institut Curie, the Anaxem platform of INRAE and the animal facility of Institut Pasteur. This work was supported by INSERM, Institut Curie, Cancéropôle Ile-de-France, Ligue contre le Cancer - comité de Paris (RS18/75-23), Fondation ARC pour la recherche sur le cancer (PJA20171206249), Cancer Research Institute (CLIP grant CRI4215), Fonds AMGEN France pour la Science et l'Humain, Institut National du Cancer (2018-1-PLBIO-01-ICR1) and Agence Nationale de la Recherche (ANR-10-LABX-0043, ANR-10-IDEX-0001-02 PSL), all received by E.S. A.De.J. is a fellow of the ERC Horizon 2020-Marie Sklodowska-Curie Actions (No 842535) and Ramón y Cajal (RyC) Programme from the Spanish State Research Agency (RYC021-031768-I). A.Co. is a fellow of Fondation pour la Recherche Médicale (FDT202001010805).

## Author contributions

Investigation: A.Co., AdJ., A.Cr., L.A., A.R., N.M., J.L.S., A.B., A.F., M.R.L., C.V., E.S. Conceptualization: A.Co., Ad.J, ES. Formal analysis: A.Co., AdJ., A.Cr., A.R., N.M., J.L.S., M.R.L., L.G., C.V., E.S. Methodology: J.L.S., L.G., S.R., C.V., E.S. Provision of essential tools: C.V., B.S., SR. Supervision: E.S. Writing – original draft and figures: AdJ. and E.S. Writing – review and editing: all authors.

## Competing interests

E.S. is an inventor on a patent entitled "Agonist of Aryl Hydrocarbon Receptor for use in cancer combination therapy" (patent WO2019057744A1). The remaining authors declare no competing interests.
