## [Transparent Peer Review file · Nature Communications]

Physiological activation of Aryl Hydrocarbon Receptor by food-derived ligands is essential for the efficacy of anti-PD1 therapy

Corresponding Author: Dr Elodie Segura

Version 0:

Reviewer comments:

Reviewer #1

(Remarks to the Author)

In this manuscript the authors employ a cancer model (injecting tumor cells subcutaneously into mice) to address the issue if anti-PD1 treatment can be enhanced by dietary supplementation with indole-3-carbinole, an AHR precursor. Given that immune checkpoint treatment is an important tool in oncology, yet fails in some instances, the authors thus address a very important topic. They hypothesize that raising the level of AHR ligands in animals via the diet can ameliorate anti-PD1 treatment failure. The manuscript first checks, if I3C feeding results in AHR-activation in the gut (this has been shown before), then move on to treat tumor-bearing mice with anti-PD1 and assess cancer growth. The next chapters are devoted to the identification of the affected cells in the tumors, and potential mechanisms, which they identify as "reinvigoration of progenitor-exhausted T cells".

While the paper is interesting, it is also hard to read due to its length and structure (two long chapters on negative results) and a certain over-interpretation of data. In particular the paper would profit from a more clear presentation of results in graphs, including clearer legends. It is also not truly rigid in presentation of AHR-modulation by the various ligands (leaving the expression that affinity is of no concern). Somewhat surprising is that ICZ - a high affinity ligand generated from dietary I3C is not mentioned or measured anywhere. They also do experiments to show that microbioata-derived AHR ligands are irrelevant in the anti-PD1 effect. This is the least convincing section, given the artificial experimental data with germ free mice or selective antibiotics. A main issue of concern for this reviewer are the small effects, which should not be over-interpreted, and a large variation in important data, which remains unaddressed (although there might be messages in there). All in all, shortening the paper might be a good idea.

Here is a - not comprehensive list - of stumbling blocks and concerns

- Include a Timeline graph of treatments, and somewhere show mice with tumors to visualize.
- The last sentence in the abstract is somewhat exaggerated

Introduction:

Give a reference for the first sentences, add more refs. for nutrition and immune system, as this is key point in your argumentation. Also, add refs to the claim that AHR ligand affect distant tissues (plural, so not just brain). Avoid or explain words such as tumor immune landscapt or cancer immune surveillance. The latter is often used in the context of gd T cells, which are not central to this manuscript.

Results.

The first chapter of the result section is of course the most important. Was the feeding the mice controlled, i.e. the two diets were equicaloric, had all the nutrients needed, and were eaten at the same rate? The Cyp-induction by a I3C diet has been done before, so this description does not need so much room. In the M&M section more info can be given, or as supplementary information. PCR data are shown as Area under the curve. This is not explained in M&M and unclear how one arrives from RT-PCR at these values. Figure 1, albeit very central to the paper remains unclear. Eg. 1A does not state how often the tumor volumes were measured, and what the starting size (80-100µm) of the individual mouse was. Only 2/6 mice fed I3C profited from the tumors, ie. the tumor size did not increase upon ant-PD1. In Fig 1B the proportion of mice

seems not a good measure given the small overall numbers.

Together the claim on Results, line 29 seems exaggerated. The figure legends can be much improved to help readers understand the graphs. Also, one would like to see comments on the very high interindividual variability, which makes statistical analysis a challenge.

page 4 line 11 - I3C should be converted in the stomach acid, Why was ICZ not measured? The axes of Figure 2 need to be equal, this helps to better compare. This also concerns the question of affinity. A high abundance of a low affinity agonist, might lead to the same AHR-activation response as a low abundance of a high affinity agonist. The authors need to take this into account in the interpretation of the statement that microbiota derived agonists are irrelevant. Page 4, line 31 ff is not easily visible from the graph and might be an overinterpretation.

pages 5 and 6. This is a matter of taste, but this reviewer thinks it is not helpful to lead the readers to a lot of negative data. This might be shortened, or put to some extent in the supplementary information, so as not to impede reading flow. In any case though, the authors should explain their rationale (and reference it) for looking at the various cells they analyze in such detail, and be less verbose. p6 line 20 - also B cells are lymphoid cells, so be more specific.

page 7 line 21 - explain why 48 hours? Can one expect a visible expansion given T cell division time and the number of responding cells in the pool.

Discussion:

The discussion is very short, and given the large set of results, and possible over-interpretation, this is unsatisfactory. They need to discuss affinity issues, pharmacokinetics, AHR-biology, the limits of their data, the high interindividual response variation, and more. They can expand on why they think the response is cancer-specific, and says something of the kind of work that still needs to be done.

referencing can be improved, eg. the impact of AHR and I3C-poor diet on gut homeostasis and GALT was first published by Veldhoen et al, and Kadow et al.

-

Reviewer #2

(Remarks to the Author)

Alba De Juan & Alice Coillard et al. report that in mice fed a diet enriched for the AHR pro-agonist indole-3-carbinol showed a stronger response to anti-PD1 blockade than mice fed a diet naturally poor in AhR ligands. Lack of dietary AhR ligands impaired NK cells infiltration upon anti-PD1 treatment. Using mice deficient for AhR in specific immune cells, the authors unraveled an essential role for AhR in CD8 T cells. RNAseq analysis revealed that AhR promotes anti-PD1-mediated reinvigoration of progenitor exhausted CD8 T cells and licenses the functional response of effector CD8 T cells including production of NK cells chemo attractants. The study provides evidence for a role of nutrients in anti-tumor immune responses and may open up new avenues for dietary interventions to improve the efficacy of immune checkpoint blockade.

The study provides evidence for the importance of eating cruciferous vegetables and elucidates the effects of their metabolites. Moreover, it sheds light on important immune functions of AHR in cancer, which appears to be ligand and context specific.

Regarding kynurenine: This may also be produced by microbiota. According to PMID 37450589, PMID: 37590143 and PMID: 30409559: Microbiota can metabolize tryptophan into N-formyl kynurenine and kynurenine.

“Collectively, these results show that AhR agonists derived from microbiota metabolism are largely dispensable for the efficacy of anti-PD1 therapy.” While the authors have used different ways to test for the role of indoles derived from microbiota, this statement still appears very strong as one could think of reasons why there might be a role that the authors were not able to capture in their experimental settings. I would hence recommend to tune down this statement a bit.

According to the Fig3D CD206+ tissue macrophages are increased upon I3C diet.

“CD206+ tissue macrophages were more abundant in tumors of mice fed on the I3C diet (Fig3D).”

“Collectively, these results indicate that lack of dietary AhR ligands does not modify significantly the baseline myeloid cell composition except for increased proportions of CD206+ tissue macrophages.”

Aren't these two sentences contradictory? I3C is the opposite of “lack of dietary AhR ligands” but in both cases the authors describe an increase of CD206+ macrophages. Shouldn't the second sentence read decrease instead of increase?

Granzyme B appears to be regulated on mRNA but not protein level, any explanation on this?

“NK cells immune surveillance”, the apostrophe is missing after cells'

“NK cell-deficient mice displayed accelerated tumor growth with larger tumors from an early stage (FigS6G), indicating that NK cells are essential for tumor control in this pre-clinical model.”

Here, I was expecting to see how the tumors of NK cell-deficient mice respond to the two different diets upon immune checkpoint blockade.

“we used a reporter mice with TdTomato fluorescence in AhR-expressing cells”, please omit “a”

“We then verified their functional...” according to Karl Popper one can only falsify not verify, I guess you examined

“To assess tumor infiltration by AhR-deficient NK cells, we inoculated MCA101-OVA tumors to Nrc1*AhRD mice and WT littermates, fed on I3C diet.” What about mice fed with indole-poor diet?

“We then inoculated MCA101-OVA tumors to Lck*AhRD mice and WT littermates, fed on I3C diet, and analyzed tumors 48h after treatment with vehicle or anti-PD1. NK cells numbers were increased in tumors upon anti-PD1 treatment, only in WT mice (Fig6E).” What about mice fed with indole-poor diet?

Figure 7 “Tumor-infiltrating 'progenitor exhausted' CD8 T cells were isolated 24h after anti-PD1 injection and analyzed by RNA-seq (n=2 independent samples).”
Using only two independent samples for this analysis is not sufficient, in particular for WT quite some variation is apparent.

To infer metabolic changes, RNASeq can be misleading. Would it be possible to confirm the results regarding oxidative phosphorylation using other methods, e.g. metabolite measurements, Seahorse or else?

CCL3 and 4 appear to be higher in the vehicle condition in indole poor diets than in I3C diets, Fig8F. Why does this not impact NK cell infiltration there?

The naming of the documents and provision of the tables as CSV files is suboptimal. Descriptive titles would be helpful. Excel tables would make it easier to obtain an overview of the supplemental tables.

Reviewer #3

(Remarks to the Author)

The authors identified a major role for diet-derived, but not microbiota-derived, AhR agonists (I3C-DIM) in boosting T and NK cell-immune responses following anti-PD1 treatment in three independent transplantable tumor models. They used very elegant experimental approaches, Td tomato tracing systems and orthogonal methods of demonstration (ATB and tryptophanase deficient E coli in germ free mice for instance, AHR deficiency driven by various promoters, sc RNA seq). This manuscript follows another piece of work dealing with AhR ligands in skin inflammation by the same authors. The paper is timely since AhR ligands harbor inconsistent effects in tumor bearers across manuscripts.

WT and LysM*AhR mice fed on I3C diet responded equally well to anti-PD1 treatment while mice fed on indole-poor diet showed limited efficacy of anti-PD1 therapy in both groups, stressing that, despite CD244 expression, mo-Mac are not the problem in mice fed with indole poor diets. Lack of dietary AhR ligands does not affect the proportion of tumor-infiltrating total or exhausted CD8 T cells, but I3C diet upregulated PD1 surface expression on CD8+T cells and Treg numbers. Migration of DC to tdLN and induction and infiltration of tumor antigen-specific CD8 T cells was not affected by the lack of dietary AhR ligands. However, impaired mitochondrial oxidative phosphorylation was altered in WT 'progenitor exhausted' T cells following anti-PD1 treatment, suggesting that AhR was required for optimal metabolic rewiring. In addition, Nr4a1 was overexpressed in AhR-deficient 'progenitor exhausted' T cells. In the last figures, they ended up redemonstrating that AhR is essential in CD8 T cell fitness, favoring the oxidative phosphorylation of progenitor exhausted Slamf6 + TIL cells and the functional cytotoxic response of CD8 effector T cells, the secretion of NK chemoattracting chemokines and the costimulatory ICOS/GITR pathways. The paper is very comprehensive, clear, nicely written with a logical flow. The supplemental data are useful in general. The originality relies on the demonstration that dietary AhR ligands impact anti-tumor immune responses in a direct manner, rather than indirectly through intestinal microbes.

Specific comments

Figure 6E (left)-G: there is decreased NK cell recruitment or infiltration in tumors of Lck* AhR mice fed by I3C diet. It would be rigorous to deplete NK cells from the WT fed with I3C and show that tumors do not respond to anti-PD1 Abs. This experiment is the only one missing from this manuscript. In the Ncr1*AhR mice fed on I3C diet and treated with anti-PD1 against MCA201-OVA, what do we observe in terms of tumor response?

What is the direct effect of AhR agonist 3,3-Diindolyl-methane (DIM), or I3AA (compared with microbial indoles I3P, indoxylsulfate) on NK cell or T cell functions (proliferation, cytotoxicity, Cxcl9 release) in vitro? Or ex vivo? In addition to Fig 7E and S8A panels.

Minor comment

Ref 54: Analyses on the set of genes in NSCLC from public data basis. Please indicate statistical analyses on the graph (* or ns) and the number of patients analysed per group in the Fig Legend.

Reviewer #4

(Remarks to the Author)

This interesting and comprehensive report focuses on the effects of the dietary indole compound I3C in anti-tumor responses to anti-PD1 therapy. The impact of the work is important, because it relates to the potential of I3C supplementation in patients to improve efficacious responses to anti-PD1 responses (although it is unclear this may also heighten autoimmune adverse effects as well). The significance of the work is also important, because it suggests that I3C-mediated AhR signaling on T cells themselves is where the positive effects of I3C are rooted. This is quite interesting, because it suggests the possibility that I3C supplementation in patients might also improve CAR-T therapy, or other types of T-cell mediated cancer immunotherapy. As a modest criticism, the benefits of I3C supplementation were somewhat limited in the model systems used (~2-fold effects): MCA101-OVA in particular that was the main model used is tuned to be extremely immunogenic, yet I3C effects were still somewhat limited, if significant. Also, the authors asserted that microbial-derived indoles can not replicate the effect of this dietary indole, but some of the data did not seem to support this assertion in a definite manner (see specific comments below). Lastly, somewhat oddly, the effects of an 'I3C agonist' called DIM appeared to be far more efficacious than I3C itself, yet the authors neither noted this nor pursued this very interesting result (see specific comments below). While not vital to the study, DIM may open a door to further studies that could heighten novelty and impact of this direction even more, and the authors might be encouraged to explore this compound more.

Specific comments

1. Figure 1. Effect of I3C diet is interesting but relatively modest, generating a two-fold increase in efficacious responses (mainly partial). The MC101-OVA model system is extremely immunogenic (engineered to be so), such that a two-fold effect is not too impressive. B16F10 is also quite immunogenic, but E0771 less so, the latter of which also shows a lesser benefit of I3C on anti-PD1 response. Lastly, the effect of I3C on the memory response triggered by anti-PD1 can not be discerned from any mouse exhibiting a complete response to anti-PD1 (as defined). Overall, while consistent, the effects of I3C appear to be fairly modest.
2. Figure 2A. Authors state that microbiota-derived AhR ligands are not affected by diet, but Fig. 2A clearly seems to show that is not the case (e.g. compare differences of I3P, I3carboxyaldehyde). Could these also be food-derived? If not, the assertion that microbiota-derived AhR ligands are not involved appears not to be uniformly supported.
3. Figure. 2I. The DIM-supplemented response to anti-PD1 is quite improved, consistent with the positive effect of I3C but even more active. This is an interesting observation: the effect is more than two-fold like I3C. If DIM an 'I3C agonist' why is it much better than I3C? Does it super-elevate I3C levels? Are there other effects of DIM beyond AhR stimulation? An experiment to compare AhR target gene activation (e.g. Cyp1a1) by I3C vs DIM would be instructive to mechanism of action here. Minimally, there should be a definition of what DIM is (what is an I3C agonist? [p.4]) and state whether it is a direct AhR ligand like I3C; and discuss the basis for the superior effect of DIM vs I3C.
4. Figures 3-7. Mechanistic studies focus on I3C instead of DIM which appears to be a far stronger stimulant of anti-PD1 efficacy. Why do the authors not pursue DIM? Can DIM be supplemented in the diet readily like I3C? From a clinical perspective, DIM looks potentially much more interesting than I3C. This should be discussed and defended in the Results section during the transition to the mechanism studies, and possibly in the Discussion, since the focus on better studied I3C seems to be a missed opportunity in terms of significance and impact.
5. Figure 6 is particularly interesting and incisive in defining a T cell-specific role for AhR signaling in modulating the anti-PD1 efficacy response. This result is a bit buried in the figure and might be highlighted earlier, although this is a minor comment. In a transplant experiment, might T cells from an I3C diet-fed mouse treated with anti-PD1 exhibit superior efficacy in a non-treated tumor bearing hosts (compared to control diet)? I wonder that these results might have implications for CAR-T therapy as well as anti-PD1 therapy, to note in the discussion perhaps.

Version 1:

Reviewer comments:

Reviewer #2

(Remarks to the Author)

The authors have addressed my questions. I am satisfied with the modifications.

Reviewer #3

(Remarks to the Author)

The authors provided all the necessary answers to most of my /our requests. I accept this version of the manuscript without any restriction.

Reviewer #4

(Remarks to the Author)

The authors provided good responses to the critique provided. Overall, I think the results, interpretation and conclusions in the revised manuscript are sound and there is sufficient methodological detail for reproduction. The findings are significant to the field of cancer immunotherapy and offer potential near-term clinical impact. I believe the work will be well-cited and will encourage clinical studies of the utility of I3C or DIM in improving anti-PD1 responses in patients (highly tractable by oral supplementation). In summary, the manuscript is noteworthy in providing what could be a simple tractable approach to improve anti-PD1 efficacy, a goal of the highest interest and significance in the field of cancer research.

Version 2:

Reviewer comments:

Reviewer #2

(Remarks to the Author)

The comments of reviewer 1 appear to be addressed.

Other points:

Rather than using the term indole-poor diet, according to Fig 2A, it appears more accurate to use the term I3C-poor diet as the other indoles are not reduced.

Some spelling should be corrected, e.g. throughout the manuscript: cells composition, or cells profiles, either an apostrophe is missing or it should read T cell composition or composition of cells. Same for cells respiration or cells isolation

Critically recheck spelling. Space missing in Legend 3.

RESPONSE TO REVIEWER COMMENTS

Reviewer #1

In this manuscript the authors employ a cancer model (injecting tumor cells subcutaneously into mice) to address the issue if anti-PD1 treatment can be enhanced by dietary supplementation with indole-3-carbinole, an AHR precursor. Given that immune checkpoint treatment is an important tool in oncology, yet fails in some instances, the authors thus address a very important topic. They hypothesize that raising the level of AHR ligands in animals via the diet can ameliorate anti-PD1 treatment failure.

In this manuscript, we aimed to address the consequence of a lack of dietary AhR ligands on the response to anti-PD1 therapy, rather than the effect of increasing levels of these ligands to improve efficacy of the treatment. Indeed, standard chow diet naturally contains phytonutrients and I3C. In a previous study, we showed that feeding mice with the purified diet AIN-93M results in significantly decreased levels of I3C in the blood compared to standard chow diet, and that supplementation of AIN-93M with 200 ppm I3C restores circulating I3C to physiological levels (<https://doi.org/10.7554/eLife.86413>). Yet, I3C levels remain lower than what is measured in mice fed with our standard chow diet, therefore this setting is not appropriate to address the effect of increased levels of dietary AhR ligands compared to a standard situation. Instead, we use this setting to address the consequence of a chronic lack of dietary AhR ligands by comparing mice fed on the indole-poor diet and mice fed on the same diet supplemented in I3C. We appreciate that this may not be the case in all studies using a similar approach, for instance some studies find increased AhR stimulation with an I3C-supplemented diet compared to standard diet (<https://doi.org/10.1016/j.cell.2011.09.025>). Therefore, to improve the understanding of our experimental setting, we have added metabolomics analysis of indole levels in the plasma of mice fed on our standard chow diet (FigS1A).

- Include a Timeline graph of treatments, and somewhere show mice with tumors to visualize.

Timelines have been added to the corresponding figures.

- The last sentence in the abstract is somewhat exaggerated

To address this comment, we have removed the word 'important' from the last sentence of the abstract.

Introduction:

Give a reference for the first sentences, add more refs. for nutrition and immune system, as this is key point in your argumentation.

It has been corrected, several references have been added.

Also, add refs to the claim that AHR ligand affect distant tissues (plural, so not just brain).

In the initial version of the manuscript, we did mention studies showing the impact of gut-derived AhR ligands on brain, skin and peritoneum. In the revised version, we added a recent reference on the role of gut-derived AhR ligands in lung (<https://doi.org/10.1038/s41586-023-06287-y>). In addition, we added references showing the wide distribution of gut-derived AhR ligands including in blood

(<https://doi.org/10.1073/pnas.0812874106>; <https://doi.org/10.1016/j.immuni.2018.08.004>)

Avoid or explain words such as tumor immune landscapt or cancer immune surveillance. The latter is often used in the context of gd T cells, which are not central to this manuscript.

We have replaced 'cancer immune surveillance' with 'immune responses against tumors', and 'landscape' with 'profile'.

Was the feeding the mice controlled, i.e. the two diets were equicaloric, had all the nutrients needed, and were eaten at the same rate?

The two diets only differ in the addition of 200 ppm of I3C powder in the I3C diet, therefore they contain the same nutrients except for I3C. Mouse weight after 3 weeks of habituation was similar between the two diet groups (see figure below), indicating that both diets were eaten at the same rate.

The Cyp-induction by a I3C diet has been done before, so this description does not need so much room. In the M&M section more info can be given, or as supplementary information.

This data is shown as supplementary information. We believe it is an important control to confirm the effect of the diet. In addition, the difference between colon and duodenum is important to show in order to interpret results related to the microbiota-derived agonists.

PCR data are shown as Area under the curve. This is not explained in M&M and unclear how one arrives from RT-PCR at these values.

We apologize for the confusion. The axis indicates A.U. for 'arbitrary units', not 'area under the curve' (which is usually abbreviated as AUC). The figure legend has been corrected for clarity.

Figure 1, albeit very central to the paper remains unclear. Eg. 1A does not state how often the tumor volumes were measured, and what the starting size (80-100 μ m) of the individual mouse was.

Mice were treated when tumors reached 80-100 mm³. This information is indicated in the text and the material and methods. Tumors were measured 3 times a week, we added this detail in the methods.

Only 2/6 mice fed I3C profited from the tumors, ie. the tumor size did not increase upon ant-PD1. In Fig 1B the proportion of mice seems not a good measure given the small overall numbers.

We apologize for the confusion. Fig1A is an example of one cohort out of 4. Fig1B summarizes results from 4 cohorts for a total of 32 mice. This is indicated in the legends. Actually, as indicated in the legend, 8 mice were used in Fig1A, and 4/8 mice responded to treatment. We have edited the figure for clarity.

Together the claim on Results, line 29 seems exaggerated. The figure legends can be much improved to help readers understand the graphs. Also, one would like to see comments on the very high interindividual variation, which makes statistical analysis a challenge.

We believe the reviewer is referring to this sentence: “Anti-PD1 treatment limited tumor progression in a majority of mice fed on I3C diet (Fig1A-B).“ Fig1B shows that 60% of the mice treated with anti-PD1 responded to treatment (when fed on I3C diet), either rejecting the tumor or showing reduced tumor growth. According to the Cambridge dictionary, a majority is “more than half of a total number”, so 60% does represent a majority.

The inter-individual variation is an integral part of the system. We have chosen purposefully to treat mice when the tumor volume reached 80-100 mm³, which induces heterogeneous responses. This mirrors the situation in human patients. This is explained in the text (page 3).

page 4 line 11 - I3C should be converted in the stomach acid, Why was ICZ not measured?

I3C can be cleaved by stomach acids, but that does not necessarily mean that all of I3C is degraded in the stomach. Indeed, I3C has been detected in the plasma in other studies (<https://doi.org/10.1158/1078-0432.CCR-04-0163>; <https://doi.org/10.7554/eLife.86413>).

I3C can give rise to a range of compounds (<https://doi.org/10.4161/cc.4.9.1993>). We did not measure all of these metabolites exhaustively. In the particular case of ICZ, we did try to measure it alongside the other metabolites. However, ICZ does not ionize well, and we were unable to identify ICZ peak in the HPLC-HRMS when using a purified standard ICZ solution. Therefore, another method may be more suitable to analyze this compound. We did not pursue this further because it was not essential to support the main findings of our manuscript.

This also concerns the question of affinity. A high abundance of a low affinity agonist, might lead to the same AHR-activation response as a low abundance of a high affinity agonist. The authors need to take this into account in the interpretation of the statement that microbiota derived agonists are irrelevant.

We thank the reviewer for pointing this out. This has been added to the discussion.

Page 4, line 31 ff is not easily visible from the graph and might be an overinterpretation.

We performed statistical analysis on all datasets. For FigS2C, there was no statistical difference between group for several metabolites including I3C, therefore we concluded on “similar abundance between groups”.

pages 5 and 6. This is a matter of taste, but this reviewer thinks it is not helpful to lead the readers to a lot of negative data. This might be shortened, or put to some extent in the supplementary information, so as not to impede reading flow. In any case though, the authors should explain their rationale (and reference it) for looking at the various cells they analyze in such detail, and be less verbose. p6 line 20 - also B cells are lymphoid cells, so be more specific.

We thank the reviewer for this suggestion. We have condensed the scRNA-seq data into one figure and removed some superfluous data that was impeding the reading flow.

page 7 line 21 - explain why 48 hours? Can one expect a visible expansion given T cell division time and the number of responding cells in the pool.

Transcriptomics analysis was performed 24h after injection of anti-PD1. We reasoned that functional effects at the protein level may take more than 24h to be detectable, therefore we analyzed tumor-infiltrating immune cells by flow cytometry at 48h.

Discussion:

The discussion is very short, and given the large set of results, and possible over-interpretation, this is unsatisfactory. They need to discuss affinity issues, pharmacokinetics, AHR-biology, the limits of their data, the high interindividual response variation, and more. They can expand on why they think the response is cancer-specific, and says something of the kind of work that still needs to be done.

We thank the reviewer for these suggestions. We respectfully disagree that the discussion section is "very short". Indeed, it represents close to 700 words. Moreover, guidelines from Nature Communications indicate that "the main text (not including Abstract, Methods, References and Figure legends) should be limited to 5,000 words" and "the Discussion should be succinct". Given that our revised text was over 5200 words, we did not expand on the discussion section. We are happy to discuss this point with the editor and modify our discussion if deemed necessary.

referencing can be improved, eg. the impact of AHR and I3C-poor diet on gut homeostasis and GALT was first published by Veldhoen et al, and Kadow et al.

We have added more references as suggested. In the case of Kadow et al, we were unable to find the article the reviewer is referring to. We would be happy to include it if the reviewer provides the PMID or DOI of this article.

-

Reviewer #2 (Remarks to the Author):

Regarding kynurenine: This may also be produced by microbiota. According to PMID 37450589, PMID: 37590143 and PMID: 30409559: Microbiota can metabolize tryptophan into N-formyl kynurenine and kynurenine.

We thank the reviewer for drawing our attention to these recent studies which had escaped our review of the literature. These references have been added.

“Collectively, these results show that AhR agonists derived from microbiota metabolism are largely dispensable for the efficacy of anti-PD1 therapy.” While the authors have

used different ways to test for the role of indoles derived from microbiota, this statement still appears very strong as one could think of reasons why there might be a role that the authors were not able to capture in their experimental settings. I would hence recommend to tune down this statement a bit.

The text has been edited. We now conclude that "these results show that decreased production of AhR agonists derived from microbiota metabolism has no detectable impact on the efficacy of anti-PD1 therapy" (page 5).

According to the Fig3D CD206+ tissue macrophages are increased upon I3C diet. "CD206+ tissue macrophages were more abundant in tumors of mice fed on the I3C diet (Fig3D)."

"Collectively, these results indicate that lack of dietary AhR ligands does not modify significantly the baseline myeloid cell composition except for increased proportions of CD206+ tissue macrophages."

Aren't these two sentences contradictory? I3C is the opposite of "lack of dietary AhR ligands" but in both cases the authors describe an increase of CD206+ macrophages. Shouldn't the second sentence read decrease instead of increase?

This is a typo. We thank the reviewer for their thorough reading. It has been corrected.

Granzyme B appears to be regulated on mRNA but not protein level, any explanation on this?

We do not have a specific explanation for this observation, but it is well described that mRNA and protein abundance do not always correlate (<https://doi.org/10.1038/s41576-020-0258-4>).

"NK cells immune surveillance", the apostrophe is missing after cells'

It has been corrected.

"NK cell-deficient mice displayed accelerated tumor growth with larger tumors from an early stage (FigS6G), indicating that NK cells are essential for tumor control in this pre-clinical model."

Here, I was expecting to see how the tumors of NK cell-deficient mice respond to the two different diets upon immune checkpoint blockade.

To better address the role of NK cells in the response to anti-PD1 therapy, we have now performed an additional experiment in which we depleted NK cells during anti-PD1 treatment in mice fed on I3C diet. This data has been added to FigS6.

"we used a reporter mice with TdTomato fluorescence in AhR-expressing cells", please omit "a"

It has been corrected.

"We then verified their functional..." according to Karl Popper one can only falsify not verify, I guess you examined

It has been corrected.

"To assess tumor infiltration by AhR-deficient NK cells, we inoculated MCA101-OVA tumors to Nrc1*AhRD mice and WT littermates, fed on I3C diet." What about mice fed with indole-poor diet?

In mice fed on indole-poor diet, NK cells (whether WT or deficient) will have low AhR activation from diet-derived agonists. In addition, anti-PD1 therapy shows limited

efficacy with this diet. Therefore this setting would be suboptimal to address the role of AhR in tumor-infiltrating NK cells. In the interest of mouse reduction, as suggested by the local ethics committee, we performed this series of experiments only with mice fed on I3C diet, a condition in which we were more likely to detect significant difference between WT and deficient mice.

“We then inoculated MCA101-OVA tumors to Lck*AhRD mice and WT littermates, fed on I3C diet, and analyzed tumors 48h after treatment with vehicle or anti-PD1. NK cells numbers were increased in tumors upon anti-PD1 treatment, only in WT mice (Fig6E).”

What about mice fed with indole-poor diet?

In mice fed on indole-poor diet, tumor-infiltrating T cells (whether WT or deficient) will have low AhR activation from diet-derived agonists. In addition, anti-PD1 therapy shows limited efficacy with this diet. Therefore this setting would be suboptimal to address the role of AhR in tumor-infiltrating T cells. In the interest of mouse reduction, as suggested by the local ethics committee, we performed this series of experiments only with mice fed on I3C diet, a condition in which we were more likely to detect significant difference between WT and deficient mice.

Figure 7 “Tumor-infiltrating 'progenitor exhausted' CD8 T cells were isolated 24h after anti-PD1 injection and analyzed by RNA-seq (n=2 independent samples).” Using only two independent samples for this analysis is not sufficient, in particular for WT quite some variation is apparent.

For this transcriptomics analysis, we had to pool tumors from 4 mice to be able to isolate 3000 viable 'progenitor exhausted' CD8 T cells. Because of this technical challenge, we only analyzed 2 independent samples. We agree with the reviewer that this is not an ideal setting. Therefore, to strengthen our results, we have provided protein or functional validation for several aspects using a greater number of biological replicates.

To infer metabolic changes, RNASeq can be misleading. Would it be possible to confirm the results regarding oxidative phosphorylation using other methods, e.g. metabolite measurements, seahorse or else?

To further address the impact of diet-derived AhR agonists on the metabolic status of tumor-infiltrating T cells, we performed flow cytometry stainings for ATP and mitochondria content in the 'progenitor exhausted' CD8 T cells. We found that mitochondria content was overall similar in both diet groups and unaffected by anti-PD1 treatment. By contrast, ATP content was increased upon anti-PD1 treatment but only in mice fed on I3C diet. These results are consistent with our findings from RNA-seq and confirm a role for dietary AhR ligands in the metabolic reactivation of progenitor exhausted T cells. These results are now shown in Figure 7F-G.

CCL3 and 4 appear to be higher in the vehicle condition in indole poor diets than in I3C diets, Fig8F. Why does this not impact NK cell infiltration there?

We agree with the reviewer that CCL3 and CCL4 basal levels are slightly higher in the tumors of mice fed on indole-poor diet, although not statistically significant. These levels are not changed upon anti-PD1 treatment, contrary to mice fed on I3C diet. Additional parameters could explain the increased presence of NK cells in the tumors of mice fed on I3C diet upon anti-PD1 treatment. In addition to chemoattraction, enhanced survival could also promote increased NK cell numbers.

The naming of the documents and provision of the tables as CSV files is suboptimal. Descriptive titles would be helpful. Excel tables would make it easier to obtain an overview of the supplemental tables.

We thank the reviewer for their suggestions. This has been modified.

Reviewer #3

Figure 6E (left)-G: there is decreased NK cell recruitment or infiltration in tumors of Lck* AhR Δ mice fed by I3C diet. It would be rigorous to deplete NK cells from the WT fed with I3C and show that tumors do not respond to anti-PD1 Abs. This experiment is the only one missing from this manuscript. In the Ncr1*AhR Δ mice fed on I3C diet and treated with anti-PD1 against MCA201-OVA, what do we observe in terms of tumor response?

To better address the role of NK cells in the response to anti-PD1 therapy, we have now performed an additional experiment in which we depleted NK cells during anti-PD1 treatment in mice fed on I3C diet. This data has been added to FigS6.

What is the direct effect of AhR agonist 3,3-Diindolyl-methane (DIM), or I3AA (compared with microbial indoles I3P, indoxylsulfate) on NK cell or T cell functions (proliferation, cytotoxicity, Cxcl9 release) in vitro? Or ex vivo? In addition to Fig 7E and S8A panels.

While these questions are of interest to the AhR biology field, we believe they are out of scope of the present study. Concerning CD8 T cells, we show (Figure 5) that they do not generally express AhR, but AhR expression is upregulated specifically in tumors. Therefore, stimulating CD8 T cells ex vivo would require first to induce AhR expression artificially, a setting that would likely be very different from the physiological situation.

Minor comment

Ref 54: Analyses on the set of genes in NSCLC from public data basis. Please indicate statistical analyses on the graph (* or ns) and the number of patients analysed per group in the Fig Legend.

It has been corrected.

Reviewer #4

This interesting and comprehensive report focuses on the effects of the dietary indole compound I3C in anti-tumor responses to anti-PD1 therapy. The impact of the work is important, because it relates to the potential of I3C supplementation in patients to improve efficacious responses to anti-PD1 responses (although it is unclear this may also heighten autoimmune adverse effects as well).

We would like to point out that standard chow diet naturally contains phytonutrients and I3C. In a previous study, we showed that feeding mice with the purified diet AIN-93M results in significantly decreased levels of I3C in the blood compared to standard chow diet, and that supplementation of AIN-93M with 200 ppm I3C restores circulating I3C to physiological levels (<https://doi.org/10.7554/eLife.86413>). Yet, I3C levels remain

lower than what is measured in mice fed with our standard chow diet, therefore we would argue that in this setting potential adverse effects are likely non-existent. We appreciate that this may not be the case in all studies using a similar approach, for instance some studies find increased AhR stimulation with an I3C-supplemented diet compared to standard diet (<https://doi.org/10.1016/j.cell.2011.09.025>) and other studies use higher I3C supplementation (for instance 1000 ppm, <https://doi.org/10.1038/s41586-023-06508-4>). Therefore, to improve the understanding of our experimental setting, we have added metabolomics analysis of indole levels in the plasma of mice fed on our standard chow diet (FigS1A).

Specific comments

1. Figure 1. Effect of I3C diet is interesting but relatively modest, generating a two-fold increase in efficacious responses (mainly partial). The MC101-OVA model system is extremely immunogenic (engineered to be so), such that a two-fold effect is not too impressive. B16F10 is also quite immunogenic, but E0771 less so, the latter of which also shows a lesser benefit of I3C on anti-PD1 response. Lastly, the effect of I3C on the memory response triggered by anti-PD1 can not be discerned from any mouse exhibiting a complete response to anti-PD1 (as defined). Overall, while consistent, the effects of I3C appear to be fairly modest.

We would like to point out that the aim of our study was to examine the impact of the lack of dietary AhR ligands on the efficacy of anti-PD1 therapy. The only difference between groups is the presence or absence of I3C in the diet. In the absence of this single compound, response to anti-PD1 treatment is impaired, with only 20% of mice responding to treatment versus 60% for the I3C diet group. We respectfully disagree that such an effect is 'fairly modest'.

2. Figure 2A. Authors state that microbiota-derived AhR ligands are not affected by diet, but Fig. 2A clearly seems to show that is not the case (e.g. compare differences of I3P, I3carboxyaldehyde). Could these also be food-derived? If not, the assertion that microbiota-derived AhR ligands are not involved appears not to be uniformly supported.

We performed statistical analysis on all datasets. For Fig2A, there was no statistical difference between groups for most metabolites including indole-3-propionic acid and indole-3-carboxyaldehyde. The graphs show overlapping distribution of datapoints, except for two outliers for indole-3-propionic acid. In the absence of statistically significant difference, we concluded that there was no major impact of the diet.

3. Figure. 2I. The DIM-supplemented response to anti-PD1 is quite improved, consistent with the positive effect of I3C but even more active. This is an interesting observation: the effect is more than two-fold like I3C. If DIM an 'I3C agonist' why is it much better than I3C? Does it super-elevate I3C levels? Are there other effects of DIM beyond AhR stimulation? An experiment to compare AhR target gene activation (e.g. Cyp1a1) by I3C vs DIM would be instructive to mechanism of action here. Minimally, there should be a definition of what DIM is (what is an I3C agonist? [p.4]) and state whether it is a direct AhR ligand like I3C; and discuss the basis for the superior effect of DIM vs I3C.

We believe there is some confusion concerning DIM. DIM is a breakdown product of I3C, therefore we used the term 'I3C-derived agonist' (p4). I3C itself is not considered an AhR ligand, but rather a precursor of AhR ligands (as indicated at the beginning of

the results section). Literature shows that I3C is chemically transformed in an acidic milieu into several products that are high affinity ligands and potent agonists of AhR (including DIM). We have edited the text for clarity. We chose the term 'pro-ligand' for I3C as suggested in a recent publication in the field (<https://doi.org/10.1016/j.bcp.2023.115626>).

4. Figures 3-7. Mechanistic studies focus on I3C instead of DIM which appears to be a far stronger stimulant of anti-PD1 efficacy. Why do the authors not pursue DIM? Can DIM be supplemented in the diet readily like I3C? From a clinical perspective, DIM looks potentially much more interesting than I3C. This should be discussed and defended in the Results section during the transition to the mechanism studies, and possibly in the Discussion, since the focus on better studied I3C seems to be a missed opportunity in terms of significance and impact.

I3C was enriched in the diet by adding I3C powder to the kibble-type AIN-93M food. DIM was supplemented by oral gavage, in the form of DIM dissolved in corn oil. It could be misleading to directly compare these results because of these differences in the experimental setting that could also impact bioavailability. The kibble-type diet was available ad libitum while the oral gavage was performed 3 times a week. Therefore, to avoid misleading interpretation of the gavage experiment, we have performed additional experiments of tumor growth in mice supplemented in I3C by oral gavage (I3C dissolved in oil, at the same concentration as DIM and IAA). Our results show that oral gavage with I3C or DIM had similar effect on the efficacy of anti-PD1 treatment in terms of tumor control. We have also edited the accompanying text for clarity.

5. Figure 6 is particularly interesting and incisive in defining a T cell-specific role for AhR signaling in modulating the anti-PD1 efficacy response. This result is a bit buried in the figure and might be highlighted earlier, although this is a minor comment. In a transplant experiment, might T cells from an I3C diet-fed mouse treated with anti-PD1 exhibit superior efficacy in a non-treated tumor bearing hosts (compared to control diet)?

We thank the reviewer for this suggestion. However, in MCA101-OVA tumors, T cells represent only around 5% of tumor CD45+ cells (see FigS6C). For example, for the transcriptomics analysis we had to pool tumors from 4 mice to be able to sort 3000 'progenitor exhausted' CD8 T cells. Therefore, purifying sufficient numbers of T cells from the tumors for a cell transfer experiment is technically too challenging. In addition, the high number of mice required for T cells purification makes it unlikely to be approved by the ethics committee. Unfortunately, we are not able to perform the proposed experiment.

I wonder that these results might have implications for CAR-T therapy as well as anti-PD1 therapy, to note in the discussion perhaps.

This is an interesting point. We added it in the discussion section.

REVIEWER #1 COMMENTS

While the paper is interesting, it is also hard to read due to its length and structure (two long chapters on negative results) and a certain over-interpretation of data. In particular the paper would profit from a more clear presentation of results in graphs, including clearer legends.

We have modified the manuscript to improve reading flow and clarity (see below for specific points).

It is also not truly rigid in presentation of AHR-modulation by the various ligands (leaving the expression that affinity is of no concern).

We have modified the discussion to include this point.

Somewhat surprising is that ICZ - a high affinity ligand generated from dietary I3C is not mentioned or measured anywhere.

I3C can give rise to a range of compounds (<https://doi.org/10.4161/cc.4.9.1993>). We did not measure all of these metabolites exhaustively. In the particular case of ICZ, we did try to measure it alongside the other metabolites. However, ICZ does not ionize well, and we were unable to identify ICZ peak in the HPLC-HRMS when using a purified standard ICZ solution. Therefore, another method may be more suitable to analyze this compound. We did not pursue this further because it was not essential to support the main findings of our manuscript.

They also do experiments to show that microbiota-derived AHR ligands are irrelevant in the anti-PD1 effect. This is the least convincing section, given the artificial experimental data with germ free mice or selective antibiotics.

We respectfully disagree with the reviewer that this series of experiments are "artificial". These are experimental models, and like all models they have some flaws. This is precisely why we used several different approaches to address this question. It would have been valuable if the reviewer would have suggested some "better" methods, but in the absence of specific suggestions, we did not perform additional experiments on this point. We did rephrase our conclusions to reflect the fact that some effects may have been missed. In any case, we did not observe a significant effect on the efficacy of anti-PD1 treatment in our model when decreasing the amount of AhR agonists produced by microbiota. This is our data and we believe it is worth reporting as it is.

A main issue of concern for this reviewer are the small effects, which should not be over-interpreted, and a large variation in important data, which remains unaddressed (although there might be messages in there).

We believe there is some misunderstanding. We have used a rather large number of individual biological replicates in all of our experiments. For instance, Fig1A is an example of one cohort out of 4. Fig1B summarizes results from 4 cohorts for a total of 32 mice per group. This is indicated in the legends.

All in all, shortening the paper might be a good idea.

We have condensed some of the negative data to shorten the results section.

Here is a - not comprehensive list - of stumbling blocks and concerns

- Include a Timeline graph of treatments,
Timelines have been added to the corresponding figures.

and somewhere show mice with tumors to visualize.

We have shown tumor volume in all of our experiments. Taking pictures of live mice is not allowed in our animal facility. The only pictures possible would be of tumors after excision. We do not understand how such images of tumors would provide more valuable information. In any case, in the interest of mouse reduction, we did not perform a new series of experiments for the only purpose of taking images after mouse sacrifice.

- The last sentence in the abstract is somewhat exaggerated

To address this comment, we have removed the word 'important' from the last sentence of the abstract.

Introduction:

Give a reference for the first sentences, add more refs. for nutrition and immune system, as this is key point in your argumentation.

It has been corrected, several references have been added.

Also, add refs to the claim that AHR ligand affect distant tissues (plural, so not just brain).

In the initial version of the manuscript, we did mention studies showing the impact of gut-derived AhR ligands on brain, skin and peritoneum. In the revised version, we added a recent reference on the role of gut-derived AhR ligands in lung (<https://doi.org/10.1038/s41586-023-06287-y>). In addition, we added references showing the wide distribution of gut-derived AhR ligands including in blood (<https://doi.org/10.1073/pnas.0812874106>; <https://doi.org/10.1016/j.immuni.2018.08.004>)

Avoid or explain words such as tumor immune landscapt or cancer immune surveillance. The latter is often used in the context of gd T cells, which are not central to this manuscript.

We have replaced 'cancer immune surveillance' with 'immune responses against tumors', and 'landscape' with 'profile'.

Results.

The first chapter of the result section is of course the most important. Was the feeding the mice controlled, i.e. the two diets were equicaloric, had all the nutrients needed, and were eaten at the same rate?

The two diets only differ in the addition of I3C powder in the I3C diet,

C57Bl/6 mice (5 weeks old at the start of the experiment) were fed on indole-poor or I3C diet for 3 weeks. Weight on the day of tumor cells inoculation (n=40). Indole-poor diet: mean=17.4, SD=0.826; I3C diet: mean=17.6, SD=0.105. Unpaired t-test: not significant.

therefore they contain the same nutrients except for I3C. Mouse weight after 3 weeks of habituation was similar between the two diet groups (see figure below), indicating that both diets were eaten at the same rate.

The Cyp-induction by a I3C diet has been done before, so this description does not need so much room. In the M&M section more info can be given, or as supplementary information.

This data is shown as supplementary information. We believe it is an important control to confirm the effect of the diet. In addition, the difference between colon and duodenum is important to show in order to interpret results related to the microbiota-derived agonists.

PCR data are shown as Area under the curve. This is not explained in M&M and unclear how one arrives from RT-PCR at these values.

We apologize for the confusion. The axis indicates A.U. for 'arbitrary units', not 'area under the curve' (which is usually abbreviated as AUC). The figure legend has been corrected for clarity.

Figure 1, albeit very central to the paper remains unclear. Eg. 1A does not state how often the tumor volumes were measured, and what the starting size (80-100 μ m) of the individual mouse was.

Mice were treated when tumors reached 80-100 mm³. This information is indicated in the text and the material and methods. Tumors were measured 3 times a week, we added this detail in the methods.

Only 2/6 mice fed I3C profited from the tumors, ie. the tumor size did not increase upon ant-PD1. In Fig 1B the proportion of mice seems not a good measure given the small overall numbers.

We apologize for the confusion. Fig1A is an example of one cohort out of 4. Fig1B summarizes results from 4 cohorts for a total of 32 mice. This is indicated in the legends. Actually, as indicated in the legend, 8 mice were used in Fig1A, and 4/8 mice responded to treatment. We have edited the figure for clarity.

Together the claim on Results, line 29 seems exaggerated.

We believe the reviewer is referring to this sentence: "Anti-PD1 treatment limited tumor progression in a majority of mice fed on I3C diet (Fig1A-B)." Fig1B shows that 60% of the mice treated with anti-PD1 responded to treatment (when fed on I3C diet), either rejecting the tumor or showing reduced tumor growth. According to the Cambridge dictionary, a majority is "more than half of a total number", so 60% does represent a majority.

The figure legends can be much improved to help readers understand the graphs. Also, one would like to see comments on the very high interindividual variability, which makes statistical analysis a challenge.

The inter-individual variation is an integral part of the system. We have chosen purposefully to treat mice when the tumor volume reached 80-100 mm³, which induces heterogeneous responses. This mirrors the situation in human patients. This is explained in the text (page 3).

page 4 line 11 - I3C should be converted in the stomach acid, Why was ICZ not measured?

I3C can be cleaved by stomach acids, but that does not necessarily mean that all of I3C is degraded in the stomach. Indeed, I3C has been detected in the plasma in other studies (<https://doi.org/10.1158/1078-0432.CCR-04-0163>; <https://doi.org/10.7554/eLife.86413>).

I3C can give rise to a range of compounds (<https://doi.org/10.4161/cc.4.9.1993>). We did not measure all of these metabolites exhaustively. In the particular case of ICZ, we did try to measure it alongside the other metabolites. However, ICZ does not ionize well, and we were unable to identify ICZ peak in the HPLC-HRMS when using a purified standard ICZ solution. Therefore, another method may be more suitable to analyze this compound. We did not pursue this further because it was not essential to support the main findings of our manuscript.

The axes of Figure 2 need to be equal, this helps to better compare.

Figure 2 has 9 panels. We do not understand which panel the reviewer is referring to. In the case of panel A for instance, the graphs show relative abundance for each metabolite, not absolute concentrations. Therefore, it is only possible to compare the two diet conditions for each specific metabolite, and it is not possible to compare values for one metabolite versus another.

This also concerns the question of affinity. A high abundance of a low affinity agonist, might lead to the same AHR-activation response as a low abundance of a high affinity agonist. The authors need to take this into account in the interpretation of the statement that microbiota derived agonists are irrelevant.

We thank the reviewer for pointing this out. This has been added to the discussion.

Page 4, line 31 ff is not easily visible from the graph and might be an overinterpretation.

We performed statistical analysis on all datasets. For FigS2C, there was no statistical difference between group for several metabolites including I3C, therefore we concluded on “similar abundance between groups”.

pages 5 and 6. This is a matter of taste, but this reviewer thinks it is not helpful to lead the readers to a lot of negative data. This might be shortened, or put to some extent in the supplementary information, so as not to impede reading flow. In any case though, the authors should explain their rationale (and reference it) for looking at the various cells they analyze in such detail, and be less verbose. p6 line 20 - also B cells are lymphoid cells, so be more specific.

We thank the reviewer for this suggestion. We have condensed the scRNA-seq data into one figure and removed some superfluous data that was impeding the reading flow.

page 7 line 21 - explain why 48 hours?

Transcriptomics analysis was performed 24h after injection of anti-PD1. We reasoned that functional effects at the protein level may take more than 24h to be detectable, therefore we analyzed tumor-infiltrating immune cells by flow cytometry at 48h.

Can one expect a visible expansion given T cell division time and the number of responding cells in the pool.

This is an interesting point. Actually, our data shows that indeed proliferating T cells can be detected 48h after injection of anti-PD1. This data is shown in Fig4B (Ki67 staining ex vivo) and Fig6G (Edu incorporation in vivo).

Discussion:

The discussion is very short, and given the large set of results, and possible over-interpretation, this is unsatisfactory. They need to discuss affinity issues, pharmacokinetics, AHR-biology, the limits of their data, the high interindividual response variation, and more. They can expand on why they think the response is cancer-specific, and says something of the kind of work that still needs to be done.

We thank the reviewer for these suggestions. We respectfully disagree that the discussion section is “very short”. Indeed, it represents close to 700 words. Moreover, guidelines from Nature Communications indicate that “the main text (not including Abstract, Methods, References and Figure legends) should be limited to 5,000 words” and “the Discussion should be succinct”. Given that our revised text was over 5200 words, we did not expand on the discussion section. We are happy to discuss this point with the editor and modify our discussion if deemed necessary.

referencing can be improved, eg. the impact of AHR and I3C-poor diet on gut homeostasis and GALT was first published by Veldhoen et al, and Kadow et al. as anti-PD1 therapy, to note in the discussion perhaps.

We have added more references as suggested. In the case of Kadow et al, we were unable to find the article the reviewer is referring to. We would be happy to include it if the reviewer provides the PMID or DOI of this article.

RESPONSE TO REVIEWERS COMMENTS

Reviewer #2 (Remarks to the Author):

Rather than using the term indole-poor diet, according to Fig 2A, it appears more accurate to use the term I3C-poor diet as the other indoles are not reduced.

We respectfully disagree with the reviewer.

Indeed, normal chow diet comprises plant components, which naturally contain indoles. Indole-3-carbinol is a well-described example, but other indoles have been shown to be present in plants, for instance indole-3-acetic acid (<https://doi.org/10.1080/19440049.2025.2459222>). Indeed, in germ-free mice fed on chow diet, we could detect in the serum both indole-3-carbinol and indole-3-acetic acid (figure S2C).

In mice fed on the purified diet, there is no plant-based component. Therefore, all indoles derived from plants are absent. This is why we termed this diet "indole-poor diet". In the study, we supplemented the purified diet with a defined dose of indole-3-carbinol. We chose indole-3-carbinol because it is well characterized in the literature, and we did not supplement with other indoles to avoid making the study too complex.

We believe that the terms "indole-poor diet" and "I3C diet" are the most appropriate, and we do not wish to change these terms.